



# Variability of greenhouse gas (CH$_4$ and CO$_2$) emissions in a subtropical hydroelectric reservoir: Nam Theun 2 (Laos PDR)

Anh-Thái Hoàng[1,2,4], Frédéric Guérin[2,3], Chandrashekhar Deshmukh[5], Axay Vongkhamsao[6], Saysoulinthone Sopraseuth[6], Vincent Chanudet[7], Stéphane Descloux[7], Toan Vu Duc[8] and Dominique Serça[1,4]

[1]Laboratoire d'Aérologie, Université de Toulouse, CNRS UMR 5560, 14 Av. Edouard Belin, 31400, Toulouse, France
[2]Géosciences Environnement Toulouse (GET), Université de Toulouse (UPS), 14 Avenue E. Belin, 31400 Toulouse, France
[3]IRD, UR 234, GET, 14 Avenue E. Belin, 31400 Toulouse, France
[4]Université de Toulouse, UPS GET, 14 Avenue E. Belin, 31400 Toulouse, France
[5]Asia Pacific Resources International Ltd., Kabupaten Pelalawan, Indonesia
[6]Nam Theun 2 Power Company Limited (NTPC), Environment & Social Division, Water Quality and Biodiversity Dept., Gnommalath Office, P.O. Box 5862, Vientiane, Lao PDR
[7]Electriciteì de France, Hydro Engineering Centre, Sustainable Development Dpt., Savoie Technolac
[8]Research of Organic Matter Group (ROOM), Environmental and Life Science Research Laboratory, Thuy Loi University, Hanoi, Vietnam

*Correspondence to*: Anh-Thái Hoàng (anh.thai2811@gmail.com)

**Abstract.** Hydroelectric reservoirs, while supporting renewable energy production, contribute notably to greenhouse gas (GHG) emissions in tropical and subtropical regions. This study presents a 14-year (2009–2022) analysis of CH$_4$ and CO$_4$ emissions from the Nam Theun 2 (NT2) reservoir, using a combination of discrete water sampling, bubbling funnel traps, and high-frequency eddy covariance (EC) measurements. Emission pathways assessed include diffusion, degassing, and ebullition. EC results of CO$_2$ fluxes were consistently higher than estimates from discrete samplings due to the system's placement in shallow, high-emission areas and its capacity to capture real-time turbulence and diurnal variability. In contrast, CH$_4$ fluxes from eddy covariance were often lower than discrete-based calculations, particularly in later campaigns, due to spatial limitations, wind filtering, and reduced sensitivity to bubbling events. The findings highlighted the importance of integrating multiple techniques to address spatial and temporal variation and reveal the influence of reservoir stratification, carbon cycling, and hydrological operations on greenhouse gas dynamics. Over the study period, total emissions reached 10736 Gg CO$_2$eq, with CH$_4$ contributing 51% (5468 Gg CO$_2$eq) and CO$_2$ 49% (5268 Gg CO$_2$eq). Emissions peaked in 2010 (1276 Gg CO$_2$eq) and declined by approximately 70% by 2021, driven by reservoir aging and depletion of labile organic matter. Seasonally, the warm dry season accounted for 3809 Gg CO$_2$eq (35.5%), the cold dry for 3841 Gg CO$_2$eq (35.8%), and the warm wet for 3086 Gg CO$_2$eq (28.7%). CH$_4$ emissions were highest in the WD season due to enhanced ebullition under stratified and anoxic conditions, while CO$_2$ emissions peaked in the CD season due to reservoir overturn and increased respiration. Ebullition dominated CH$_4$ emissions (77%) and remained stable over time, whereas diffusive CH$_4$ fluxes declined by 97%. CO$_2$ emissions were majorly diffusive (96%) and showed consistent decline (-87%). This long-term dataset improved the understanding of subtropical reservoir emissions and provided insights for global carbon budgets and improving the climate impact assessment of hydropower development.

## 1 Introduction

Hydroelectric reservoirs are a cornerstone of the global renewable energy portfolio, with the number of large dams expanding rapidly, especially in tropical/subtropical region (Zarfl et al., 2015). By December 2024, according to The International Commission On Large Dams, of 62339 dams registered worldwide, approximately 17% of them are used for electricity



40 generation (10567 dams). While hydro-reservoirs are often regarded as contributors to renewable and green energy production, their net impact on the carbon cycle and GHG emissions remains uncertain. Evidence suggests that they may also serve as significant sources of GHGs, highlighting their role in the context of climate change (Barros et al., 2011; Deemer et al., 2016; Prairie et al., 2018; St. Louis et al., 2000; Yan et al., 2021). Large quantities of methane ($CH_4$) and carbon dioxide ($CO_2$) are emitted primarily through microbial decomposition of organic matter (OM) in both the water column and the flooded OM at

45 the bottom of the reservoir. At the global scale, Barros et al. (2011) estimated that hydroelectric reservoirs release approximately 100 Tg $CO_2$-equivalent ($CO_2$eq) of $CH_2$ and 176 Tg $CO_2$eq of $CO_2$ annually. Higher estimates have also been reported by Hertwich, (2013), who suggested annual emissions of 243.3 Tg $CO_2$eq of $CH_4$ and 278 Tg $CO_2$eq of $CO_2$. The highest estimate of greenhouse gas emissions from reservoirs was reported by St. Louis et al., (2000) who calculated annual emissions of 1833.3 Tg $CO_2$eq year$^{-1}$ for $CH_4$ and 990 Tg $CO_2$eq year$^{-1}$ for $CO_2$. These emissions are especially pronounced in

50 tropical and subtropical regions, where warm temperatures and abundant organic inputs enhance methanogenesis and overall OM decomposition rates (Barros et al., 2011).

 $CH_4$ production occurs predominantly under oxygen-depleted conditions during the mineralization of flooded organic matter from soil and vegetation. According to Deemer et al. (2016), $CH_4$ emissions constitute approximately 80% of total $CO_2$eq emissions from reservoir water surfaces, a percentage which could reach up to 90% when calculated over a span of 100 years.

55 The primary emission pathways include ebullition, diffusive fluxes at the surface of the reservoir, and degassing downstream of the turbines. Disregarding degassing, ebullition is the dominant mechanism and contributes about 65% of total $CH_4$ emissions, while diffusion accounts for the remaining 35% (Deemer et al., 2016).

 $CO_2$ emissions from reservoirs arise primarily from the supersaturation of $CO_2$ at the water surface. This supersaturation results from heterotrophic respiration driven by terrestrial carbon inputs from the watershed and flooded OM (Abril et al., 2005). $CO_2$

60 from reservoirs is predominantly emitted to the atmosphere through diffusive fluxes and degassing (Deshmukh et al., 2018). Emissions of $CO_2$ can vary significantly daily (day-night) (Liu et al., 2016; Podgrajsek et al., 2015) and seasonnally (Morales-Pineda et al., 2014) scales, and episodically, in response to weather-induced hot moments (Liu et al., 2016).

 GHG fluxes from reservoirs exhibit considerable temporal as well as spatial variability (Colas et al., 2020; Guérin et al., 2016). This variability is influenced by multiple factors, including fluctuations in water discharge, variations in carbon and nutrient

65 inputs from the watershed, changes in water depth, and seasonal shifts in temperature (Abril et al., 2005; Linkhorst et al., 2020). Therefore, acquiring representative measurements at high spatial and temporal resolution is crucial for accurately estimating emissions at the reservoir scale.

 The NT2 reservoir in Laos offers a valuable case study for examining GHG flux temporal dynamics and spatial variability in subtropical hydroelectric systems. Over a 14-year monitoring period, NT2 has demonstrated clear seasonal and interannual

70 trends in $CH_4$ and $CO_2$ emissions (Deshmukh et al., 2018; Guérin et al., 2016; Serça et al., 2016). A combination of advanced measurement techniques, including EC and discrete sampling, has been employed to capture high-resolution emission data. Discrete sampling is limited in its ability to capture diurnal and episodic flux variability driven by weather events (Liu et al., 2016; Podgrajsek et al., 2015). In contrast, the EC method allows for continuous measurement of $CO_2$ exchange between the



water surface and the atmosphere over a broad area, typically several hectares—without disturbing the air–water interface (Erkkilä et al., 2018). This makes EC particularly useful for capturing short-term variability and improving spatial and temporal resolution in GHG flux assessments. The approach combining these two types of approach enables the identification of key emission pathways such as ebullition and diffusion, which are influenced by physical mixing, biological activity, and water chemistry. This study aims to investigate the seasonal and long-term trends (2009-2022) of $CH_4$ and $CO_2$ emissions from the NT2 reservoir, providing a comprehensive analysis of their temporal dynamics and associated emission pathways. By integrating high-resolution EC measurements with dissolved gas analysis, this research offers valuable insights into the comparison of these two approaches and examines the key factors regulating GHG fluxes. The findings contribute to improving carbon budget assessments for subtropical reservoirs and enhancing our understanding of their role in regional and global greenhouse gas emissions.

## 2 Materials and methods

### 2.1 Study site

The Nam Theun 2 Reservoir (Figure 1), located atop the Nakai Plateau in the Khammouane Province of Lao PDR, was impounded in April 2008. It attained its first maximum water level of 538 m above sea level (asl) in October 2009, and was commissioned in April 2010. Detailed descriptions on the main characteristics of this trans-basin hydroelectrical project are provided in Descloux et al., (2016). The construction of the reservoir resulted in the flooding $5.12 \pm 0.68$ Mt C of various types of land-cover, notably forests, agriculture lands and wetlands (Descloux et al., 2011).

Situated within the sub-tropical climate zone of Northern Hemisphere (17°59'49"N 104°57'08"E) and influenced by monsoons, the reservoir experiences three distinct seasons, namely the cold dry season (CD, from mid-October to mid-February), the warm dry season (WD, from mid-February to mid-June), and the warm wet season (WW, from mid-June to mid-October). Regardless of the season, daytime is considered to start at 6:30 and end at 18:30 daily.



95

**Figure 1: Map of the Nam Theun 2 Reservoir (Lao People's Democratic Republic) with routine monitoring stations and occasional EC stations; see detail on stations in section 2.2.**

Over the study period from 2009 to 2022, the seasonally averaged air temperature was lowest during the CD season (19.6°C), whereas the warm seasons exhibited relatively stable temperatures, with 23.9°C in WD and 23.6°C in WW (Figure 2: Monthly





averages of Air temperature (A), Windspeed (B), reservoir surface area (C), Discharges (D), and Annual total rainfall (E). In the first 3 graphs, black dots represent the EC campaigns. In graph D, line represents input discharges, dot line represents outputs to the powerhouse and dot-dash line represents the spillway and dam release. In graph E, grey represents WD, white represents WW, and hatched represents CD.A). The wind speed at the reservoir by ERA5 reanalysis (Muñoz-Sabater, 2019) was consistently low, with values ranging from 0.34 to 5.19 m s$^{-1}$. Specifically, the WD exhibited an average wind speed of 1.42 m s$^{-1}$, while the WW had average wind speed of 1.28 m s$^{-1}$. In contrast, the CD experienced a relatively higher average wind speed of 2.04 m s$^{-1}$ (Figure 2B). The annual average rainfall recorded at the site was 2445 mm, with approximately 80% of the rain events occurred in the WW season (Deshmukh et al., 2014; Guérin et al., 2016). The years with the highest rainfall were 2011 (3162 mm) and 2018 (3302 mm), while the driest years were 2020 (1706 mm) and 2022 (1802 mm) (Figure 2E).

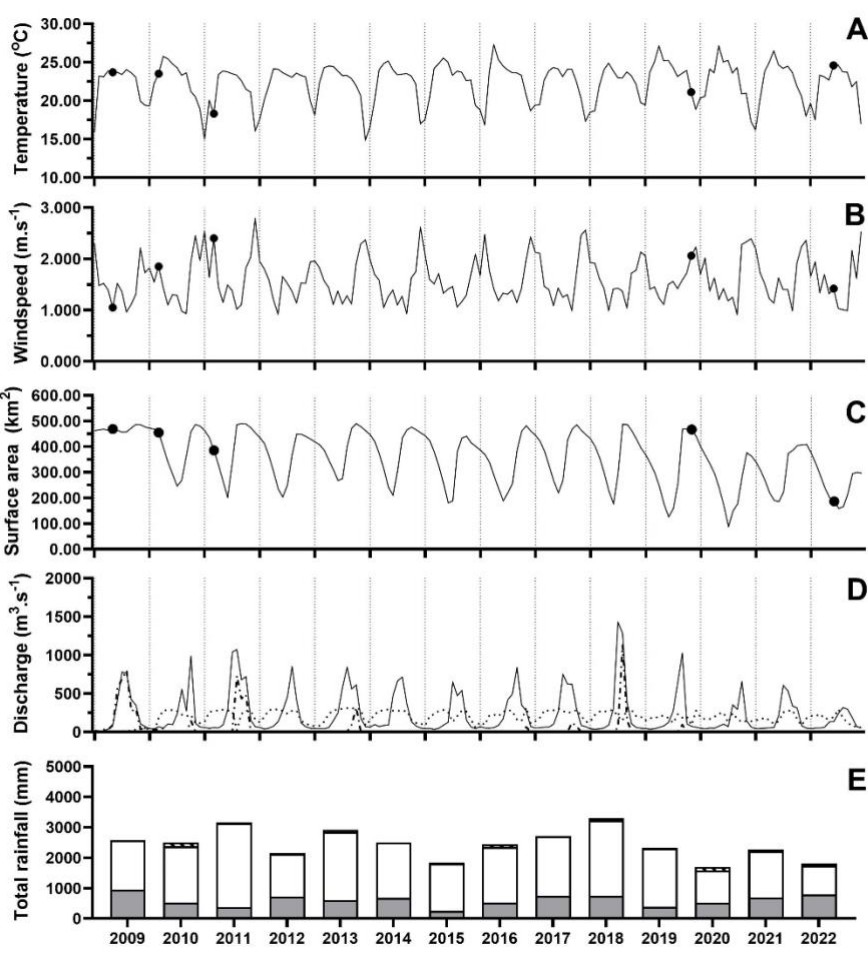

**Figure 2: Monthly averages of Air temperature (A), Windspeed (B), reservoir surface area (C), Discharges (D), and Annual total rainfall (E). In the first 3 graphs, black dots represent the EC campaigns. In graph D, line represents input discharges, dot line represents outputs to the powerhouse and dot-dash line represents the spillway and dam release. In graph E, grey represents WD, white represents WW, and hatched represents CD.**



Due to the significant variations in hydrology, the water level changed between 525.31 m asl and 538.36 m asl during the
course of study, with the lowest average water level in the WD season (532.41 m asl) and the highest one in the CD (536.20
m asl). Therefore, the reservoir surface area ranged between 81.96 km$^2$ and 499.07 km$^2$, with similar seasonal pattern to the
water level variation (Figure 2C). Categorized as shallow, the reservoir had an average depth of 7.8 m (Deshmukh et al., 2014).
On a daily basis, and for the 14 years of measurements (2009-2022), the reservoir received an average input discharge of 238.2
m$^3$ s$^{-1}$ from upstream almost pristine rivers with large seasonal variations. During the warm wet (WW) season, the average
daily input was 533.24 m$^3$ s$^{-1}$. This inflow decreased progressively from CD season, where it stood at 100.47 m$^3$ s$^{-1}$, to the WD
season, where it decreased down to 79.74 m$^3$ s$^{-1}$. The water has been permanently discharged to the downstream of the reservoir
through the dam, the spillway and to the powerhouse. The spillway release was occasionally used from October 2009 to April
2010, i.e. before the complete operation of the hydropower plant. Since then, it has been only used to flush the occasional flood
from the reservoir during raining seasons (Figure 2D). The release at the dam to the Nam Theun River was approximately 2.0
m$^3$ s$^{-1}$ on daily average and remained more or less constant (minimum release flow). Discharge to the powerhouse or output
discharge started with commissioning in April 2010, and was on average equal to 208.40 m$^3$ s$^{-1}$ (Figure 2D).

## 2.2 Sample strategies

Nine monitoring stations (RES1 to RES9) across the reservoir area, one station downstream of the turbine, at the inlet of the
artificial downstream channel (DCH1), and three station on Nam Kathang rivers (NKT1, 2, 3) were set up to collect physical
parameters (water temperature and dissolved oxygen concentration), dissolved concentration of CH$_4$, and carbon species (total
inorganic carbon IC, dissolved organic carbon DOC, and particulate organic carbon POC) in the water column. In the reservoir,
between three and six water samples were collected along the vertical profile, from the surface to the bottom, depending on
the presence of stratification features such as the thermocline and oxycline. In-situ parameters were recorded at one-meter
intervals throughout the water column. By contrast, at river and downstream channel stations, only surface water samples were
obtained (Chanudet et al., 2016).

From January 2009 until April 2017, the sampling was performed weekly to fortnightly. Afterwards, the number of stations
was reduced (RES2, RES6 and RES7 was discontinued) and the sampling frequency was degraded to monthly. The sampling
resolution of the stations RES3, RES5 and RES8 was also affected and only surface samples were gathered. Details on the
characteristics of each station can be found in Descloux et al., (2016) and Guérin et al., (2016). Notably RES1 was 100 m
upstream of the Nakai Dam and RES9 was located 1 km upstream of the water intake.

CH$_4$ ebullition was collected using the submerged funnel technique (Deshmukh et al., 2014) during three periods: from May
2009 to June 2011 through five different field campaigns, from March 2012 to September 2013, with a weekly sampling
frequency, and from March 2014 to September 2022, with a monthly sampling frequency. During the first two periods, samples
were gathered at seven stations, covering various types of flooded land-cover, namely dense, medium, light and degraded
forests, as well as agricultural soils. From 2014 the number of stations was reduced to three, covering dense and degraded



forests along with agricultural soil. The depth of each sampling sites ranged between 0.4m to 16m since no ebullition occur deeper.

The EC system was deployed in five field campaigns: a 7-day deployment in May 2009, at the transition from the WD to the WW season; a 14-day campaign in March 2010; and a 5-day campaign in March 2011 at the transition between the CD and

the WD season (Deshmukh et al., 2014). Additionally, a 24-day measurement campaign was conducted in November and December 2019, during the CD season, and a 12-day campaign in June 2022, at the transition between the WD and the WW season. The first three EC campaigns were located near the RES8 station (17°41.56'0 N, 105°15'36'0 E), with a tower installed in the middle of the reservoir surface. As a matter of consequence, the EC footprint area was homogeneous in all wind direction with the same water column depth. The latter two deployments occurred near the RES4 station (17°48'26.8"N, 105°03'27.9"E).

The tower was then located on the shore of the reservoir, approximately 1 m from the shoreline and thus, occasionally captured fluxes from the terrestrial ecosystems (see QC/QA on EC fluxes).

Air temperature, wind speed and direction data were obtained from the ERA5-Land reanalysis dataset (Muñoz-Sabater, 2019), with data provided at hourly intervals. Daily rainfall data were recorded by rain gauges located at Nakai dam (close to RES1). Daily water level measurements were conducted at Thalang station (RES4). Using the water level data in conjunction with the

reservoir capacity curve (NTPC, 2005), daily area and volume calculations of the reservoir were performed. Precise daily outflow measurements from the reservoir were taken at two points: Nakai Dam (RES1) and the Powerhouse (RES9). Additionally, in the regulation pond downstream, daily outflow to the downstream channel (DCH1), inflows and outflow of the Nam Kathang River were also monitored. Total daily inflows to the reservoir, including tributaries and rainfall, were determined through a mass balance approach. This calculation considered changes in water volume and monitored outflows

(Chanudet et al., 2012).

## 2.3 Experimental methods

Water temperature and dissolved oxygen concentration were measured in situ using a Quanta® multi-parameter probe (Hydrolab, Austin, Texas) with vertical resolution intervals of 0.5 m for the upper 5 m, and 1 m below, extending to the bottom of the sampling site.

Additionally, for laboratory analysis, water samples were continuously collected since 07/01/2009. Surface samples were acquired using a surface water sampler (Abril et al., 2007), while samples from various depths in the water column were collected using a Uwitec water sampler. Depending on the station depth, three to six samples were collected from the surface to the oxic-anoxic interface and down to the bottom of the stations.

For dissolved $CH_4$ concentration, the headspace method (Guérin & Abril, 2007) was conducted by collecting two 60 mL glass

vials (2 replicates) at each depth. To ensure the absence of bubbles in the water, a water flow with a volume at least three times that of the vial was flow through it. Subsequently, the vial was carefully capped using a butyl cap and secured with aluminium crimps. The vials were then treated with 0.3 mL of mercury chloride (1 mg $L^{-1}$). In the laboratory, approximately 30 mL of headspace was created by injecting $N_2$ to the vials. Subsequently, the vial was strongly shaken to ensure gas-liquid equilibrium.



The vial, then, was allowed to stabilize at room temperature (25°C) for at least one hour, and stored to be analysed within 15
days. $CH_4$ dissolved concentrations in the headspace were determined using gas chromatography (GC) with a flame ionization
detector (FID) (SRI 8610C gas chromatograph, Torrance, CA, USA). Each injection into the GC necessitated 0.5 mL of gas
collected from the headspace. Calibration was conducted using commercial $CH_4$ gas standards of 2 ppmv, 10 ppmv, and 100
ppmv, in a mixture with Nitrogen ($N_2$). Duplicate injections of samples exhibited a reproducibility rate higher than 5%
(Deshmukh et al., 2018; Guérin et al., 2016). Finally, to determine the dissolved concentration, the $CH_4$ gas solubility, as a
function of temperature and salinity, as described by Yamamoto et al., (1976), was used. A salinity value of 0 was assumed in
all samples.

For carbon species, filtered (0.45 μm, Nylon) and unfiltered samples were collected and analysed using a Shimadzu TOC-
VCSH analyser for IC and total organic carbon (TOC). POC was then determined by subtracting the DOC concentrations from
the TOC measurements (Deshmukh et al., 2018).

**2.4 Calculation of dissolved $CO_2$ concentration**

The dissolved $CO_2$ concentrations were computed using the CO2sys model (Lewis & Wallace, 1998), which relied on the
carbonic acid dissociation constants found in Millero (1979) for freshwater, along with the $CO_2$ solubility constants from Weiss
(1974). IC and pH using the NBS scale, together with water temperature and salinity, were employed to estimate the dissolved
$CO_2$ concentration from every water sample collected. the salinity value of 0 was assumed.

**2.5 Total diffusive flux calculation**

The $CH_4$ and $CO_2$ diffusive fluxes were computed based on the surface dissolved gas concentrations and the thin boundary
layer (TBL) equation. These fluxes were determined by the gradient of gas concentrations at the water-air interface with:

$$F = k_T \times \Delta C \qquad (1)$$

Where F represents the diffusive flux at the water-air interface, $\Delta C$ denotes the concentration gradient between the water and
the air and $k_T$ stands for the gas transfer velocity at temperature (T) with:

$$k_T = k_{600} \times (600/Sc_T)^n \qquad (2)$$

in which $Sc_T$ represents the Schmidt number of the gas at temperature (T) (Raymond et al., 2012). The exponent n was
determined based on the wind intensity on the day of measurement, taking a value of either -2/3 for wind speeds below 3.7 m
$s^{-1}$, and -1/2 for higher wind speeds (Jähne et al., 1987).

For $k_{600}$ within the reservoir, we combined the two $k_{600}$ equations from Guérin et al., (2007) and MacIntyre et al., (2010) in
order to consider both the effect of windspeed (m $s^{-1}$) and rainfall (mm $h^{-1}$) (Deshmukh et al., 2014; Guérin et al., 2016). At
station RES9, located upstream of the water intake, a constant value of 10 cm $h^{-1}$ was assigned to $k_{600}$. This decision was based
on the particular turbulence (eddies, water current) observed in this area (Guérin et al., 2016) due to the strong water
withdrawing.





To compute the total diffusive flux from the reservoir, the total area was attributed to the nine stations. Regarding the physical model presented in Chanudet et al., (2012), RES9 was handled separately due to its unique hydrological and hydrodynamic attributes. A fixed area of 3 km², consistent throughout the study period, was allocated to RES9. Similarly, RES3 located in the middle of the flooded forest, representing an separate area of the reservoir, constituting 5.5% of the total area when at full water level, was treated independently, unaffected by temporal variations (Chanudet et al., 2012).

A series of statistical tests were conducted using GraphPad Prism (GraphPad Software, Inc.) to ascertain the possibility of grouping the remaining seven stations for the area after excluding RES3 and RES9. Initially, a normality test was conducted to assess whether the data followed a normal or lognormal distribution. Subsequently, Kruskal-Wallis and Mann-Whitney tests were employed to explore similarity. The analysis confirmed that there were no significant differences between the fluxes observed at the seven stations regarding spatial variation. Consequently, the fluxes at the stations RES1, RES2, RES4, RES5,

RES6, RES7, RES8 were averaged all together for the calculation of the total diffusive emission.

## 2.6 Total degassing calculation

The degassing is assessed by the difference of concentrations of dissolved gases (C) upstream and downstream of the degassing structure. This difference was then multiplied by the corresponding discharge (Galy-Lacaux et al., 1997) with:

$$\text{Degassing} = (C_{upstream} - C_{downstream}) \times \text{discharge rate} \qquad (3)$$

The first degassing site is located downstream of the Nakai Dam. Notably, in the dam's design, the release gate was positioned within the epilimnion of the water column. Consequently, for the upstream component, concentrations from the surface to a depth of 10 m at RES1 were averaged. The surface concentration at NTH3 (Nam Theun River) was considered for the downstream component.

The second degassing site is located downstream of the powerhouse, specifically in the upper section of the downstream

channel. At this site, the balance accounted for inputs from RES9 and the Nam Kathang River (sampling points NKT1 and NKT2), and outputs discharging into the downstream Nam Kathang River (NKT3) and the downstream channel (DCH1). Since the water is permanently mixed over the whole profile at RES9, the average concentration from surface to bottom was used there, while surface concentrations were used for the other stations.

## 2.7 Gap-filling method

Due to technical difficulties such as GC failure, as well as unexpected circumstances like the Covid-19 pandemic and closure of the reservoir for maintenance, numerous data gaps were observed where no sampling were obtained for an entire month at a station. To address this issue, a gap-filling method was applied based on seasonal averaging. Each season (based on the three distinct seasons - WD, WW, CD - in a year previously mentioned) comprising four months. In cases where one, two, or three months of data were missing within a season, the available data from the remaining month(s) of that same season were used

to compute a seasonal average. This average was then used to fill the missing values for the corresponding month(s), ensuring consistency and minimizing bias in the seasonal estimates.



## 2.8 Total bubbling calculation

An artificial neural network (ANN) approach was performed (similar to what had been described by Deshmukh et al., 2014) to compute the seasonal and annual $CH_4$ ebullition flux from individual flux measurements (submerged funnels). The inputs included time series of total static pressure, change in total static pressure, and bottom temperature associated with the measured fluxes.

## 2.9 Gross emission (GE) calculation in CO₂ equivalents (CO₂eq)

$CH_4$ emission components were converted to $CO_2eq$ using a factor of 27, which represents the global warming potential (GWP) of non-fossil methane relative to $CO_2$ over a 100-year period, as defined by the IPCC (2021). Subsequently, the flux from the different pathways were summed to determine the total emissions for each gas, as well as the overall gross emissions of the reservoir, categorized by seasons and years.

## 2.10 EC system setting and data processing

The calculation of $CH_4$ and $CO_2$ EC fluxes involved assessing the covariance between scalar variables and vertical wind speed fluctuations, complying to the well-established protocols (Aubinet et al., 2001). Fluxes were positive when indicating fluxes from the water surface to the atmosphere, and negative when from the reverse direction (Deshmukh et al., 2014).

The EC set-up comprised a 3D sonic anemometer, specifically the Windmaster Pro (Gill Instruments, Lymington Hampshire, UK) used during the first two field campaigns in May 2009 and March 2010, and a CSAT-3 (Campbell Scientific, Logan, UT, USA) employed during the last three campaigns (in 2011, 2019 and 2022). Additionally, for all the campaigns, an open-path $CO_2/H_2O$ infrared gas analyser (LI-7500, LI-COR Biosciences, Lincoln, NE, USA) and a closed-path fast $CH_4$ analyser (DLT100, FMA from Los Gatos Research, CA, USA), were used. Air was carried to the DLT-100 through a 6 m-long tube (Synflex-1300 tubing; Eaton Performance Plastics, Cleveland), featuring an internal diameter of 8 mm. Positioned 0.20 m behind the sonic anemometer, the tube inlet was shielded with a plastic funnel to prevent ingress of rainwater. Furthermore, an internal 2 μm Swagelok filter was used to safeguard the sampling cell against dust, aerosols, insects, and droplets. High-frequency air sampling was achieved through the use of a dry vacuum scroll pump (XDS35i, BOC Edwards, Crawley, UK), delivering a flow rate of 26 L min$^{-1}$. Data acquisition was facilitated by a Campbell datalogger (CR3000 Micrologger®, Campbell Scientific). Given the remote location of our study site during the first three campaigns, a 5 kVA generator running on gasoline provided power for the entire EC instrumentation setup. From 2019, the power was provided directly from a domestic facility close to the site. To ensure the atmospheric $CH_4$ and $CO_2$ concentration measurements were coming from solely the water body, tests were conducted using wind direction and a footprint model (Kljun et al., 2004).

During each EC deployment, parameters such as wind speed, atmospheric pressure, air temperature, relative humidity, and rainfall were measured using a WXT 510 device (Vaisala, Finland). In addition, incoming and outgoing shortwave and longwave radiations were measured by a radiometer (CNR-1, Kipp & Zonen, Delft, The Netherlands).



The $CH_4$ and $CO_2$ EC fluxes were calculated from the 10 Hz raw data file using EdiRe software (Clement, 1999) for the first three campaigns from 2009 to 2011 (see detailed setting in Deshmukh et al., 2014) and EddyPro software for the last 2

campaigns (version 7.07, LI-COR). The timestep used in both cases was 30-minute average.

For the EddyPro setup, while generally similar to the EdiRe configuration, several methodological adjustments were made at each processing step. Initially, a spike removal procedure was applied to identify and eliminate outlier data, allowing for a maximum spike occurrence of 5% (Vickers & Mahrt, 1997). Secondly, a tilt (coordinate) correction was employed using the double rotation method. Thirdly, frequency response loss corrections were implemented to address flux losses at both low and

high frequencies, including the application of high-frequency correction factors to account for losses coming from inadequate sampling rates (Moncrieff et al., 1997). Fourthly, time lag compensation for close-path DLT-100 $CH_4$ analyser was enabled. Fifthly, the turbulence fluctuations were calculated using block-averaging method, which involved determining the mean value of a variable and assessing turbulence fluctuations by measuring deviations of individual data points from this mean (Gash & Culf, 1996). Lastly, the compensation of density fluctuation by Webb-Pearman-Leuning density correction (Webb et al., 1980)

was used only for $CO_2$ flux calculation.

## 2.11 Quality control of EC fluxes

A comprehensive set of criteria was employed to determine the acceptance or rejection of fluxes, as described in Deshmukh et al., (2014). Fluxes were tested for non-stationarity based on the methodology proposed by Foken & Wichura (1996) where fluxes were deemed acceptable only if the difference between the mean covariance of sub-records (5 minutes) and that of the

entire period (30 minutes) fell below 30%. Then, fluxes were discarded if their intermittency surpassed 1, in accordance with the criteria established by Mahrt (1998). Thirdly, ensuring the vertical wind speed component remained within specified bounds, skewness and kurtosis were utilized, following the guidelines of Vickers & Mahrt (1997), with values constrained to the ranges of (-2, 2) and (1, 8) respectively. Furthermore, the momentum flux, u'w' was mandated to exhibit negativity, signifying a downward-directed momentum flux attributed to surface friction. Moreover, fluxes were invalidated in cases

where wind originated from the power generator unit (2009-2011) or coming from the land (2019, 2022), in line with the footprint model presented by Kljun et al. (2004), considering variations in footprint extension and prevalent wind directions across diverse field campaigns. Moreover, a systematic quality check by EddyPro followed the flagging policies of Foken et al., (2005) and Mauder & Foken (2006) were also applied.

The acceptance rate for $CH_4$ fluxes in the first three campaigns was reported at 57% (59% for daytime and 52% for nighttime

fluxes; Deshmukh et al., 2014). For $CO_2$ fluxes, the application of quality control criteria resulted in the acceptance of 39% of the flux data (38% for daytime an 40% for nighttime fluxes). This proportion of validated data was similar to earlier studies documenting EC measurements conducted over lakes (Erkkilä et al., 2018; Liu et al., 2016; Mammarella et al., 2015; Morin et al., 2018; Podgrajsek et al., 2015; Shao et al., 2015).





Due to the instability of measurement conditions, including electrical disturbances encountered during the later campaigns, the
removal rates for both gases increased significantly. Therefore, the diurnal variation (day vs. night measurements) was
analysed only with the first three campaigns. In the 2019 campaign, while 42% of the original $CO_2$ fluxes were retained
following flag analysis, only 11,8% of $CH_4$ fluxes remained after QC analysis. During the 2022 campaign, both $CO_2$ and $CH_4$
fluxes demonstrated similarly low acceptance rates of 12.5% after all quality control measures applied. However, these values
are consistent with the order of magnitude reported in a recent study using EC techniques for a reservoir (Hounshell et al.,
2023), which faced similar issues related to wind direction filtering, instrument maintenance, and power instability.

## 2.12 Upscaling EC fluxes and comparison with corresponding discrete sampling

EC fluxes after quality control, were assumed to represent the flux from the entire reservoir. Accordingly, total monthly
emissions for each campaign were estimated by upscaling the fluxes using the reservoir surface area. These upscaled emissions
were then compared with the GE, excluding downstream degassing, calculated for the corresponding month of each campaign.

## 2.13 Statistical analysis

In addition to the specific statistical tests described in section 2.5, group differences were evaluated using either a t-test or
analysis of variance (ANOVA) in GraphPad Prism. The selection between parametric and non-parametric tests (such as the
Mann-Whitney or Kruskal-Wallis tests, which compare median values) was based on the distribution characteristics of the
data sets. The Kolmogorov–Smirnov test was used to assess data normality.

Standard deviation (SD) and standard error (SE) for each data set were calculated in Microsoft Excel 2019 using the following
equation:

$$SE = SD \times n^{-1/2} \tag{4}$$

## 3. Results

## 3.1 Assessment of EC data compared to GE from the reservoir's water surface

$CH_4$ fluxes from the first three campaigns (Deshmukh et al., 2016), determined by EC, ranged from 0.18 to 26.84 mmol m$^{-2}$ d$^{-1}$ and average between $5.80 \pm 0.43$ to $7.20 \pm 2.90$ mmol m$^{-2}$ d$^{-1}$. At the end of the study period, the upper end of the flux range
decreased to 25.74 mmol m$^{-2}$ d$^{-1}$ in 2019 and 13.72 mmol m$^{-2}$ d$^{-1}$ in 2022 as compared to the 2009-2011 period. Consequently,
the average fluxes also declined in 2019-2022 (Table 1), especially in 2019, which had a high number of low fluxes, resulting
in an average of $1.50 \pm 0.32$ mmol m$^{-2}$ d$^{-1}$.

2010 showed the most significant difference between day and night emissions, with daytime $CH_4$ fluxes being 1.9 times higher
than nighttime fluxes, measured at 6.95 mmol m$^{-2}$ d$^{-1}$ and 3.63 mmol m$^{-2}$ d$^{-1}$, respectively. In contrast, in 2011 the daytime



fluxes were only 1.3 times higher than nighttime values, recorded at 8.80 mmol m$^{-2}$ d$^{-1}$ and 6.30 mmol m$^{-2}$ d$^{-1}$, respectively. In 2009, despite the low acceptance ratio of fluxes (Table 1), a similar trend of higher daytime emissions remained, with a 1.7-time increase in daytime fluxes over nighttime, measured at 7.57 mmol m$^{-2}$ d$^{-1}$ and 4.37 mmol m$^{-2}$ d$^{-1}$, respectively.

**Table 1: Eddy covariance CH$_4$ and CO$_4$ flux data (mmol m$^{-2}$ d$^{-1}$) for five field campaigns., n: Number of Measurements**

|  | CH$_4$ |  | CO$_2$ |  |
|---|---|---|---|---|
|  | **Range** | **Average ± SE (n)** | **Range** | **Average ± SE (n)** |
| May - 2009 | 2.07 – 16.16 | 6.50 ± 0.53 (39)[1] | 34.06 – 493.15 | 140.00 ± 11.81 (53) |
| March – 2010 | 0.18 – 26.84 | 5.80 ± 0.43 (138)[1] | 12.95 – 616.25 | 176.41 ± 10.28 (175) |
| March – 2011 | 2.85 – 16.85 | 7.20 ± 2.90 (105)[1] | 11.00 – 169.75 | 73.16 ± 3.84 (98) |
| November – 2019 | 0.01 – 25.74 | 1.50 ± 0.32 (103) | -240.21 – 331.21 | 69.09 ± 4.43 (377) |
| June - 2022 | 0.17 – 13.72 | 4.90 ± 0.50 (61) | -67.07 – 247.10 | 98.07 ± 9.61 (44) |

[1]Data published by Deshmukh et al., 2016

The reservoir acted as a source of CO$_2$ in the early years (2009-2011), with 30-minute fluxes varying from 11.00 to 616.25 mmol m$^{-2}$ d$^{-1}$ (Table 1). On the other hand, during the last two campaigns, the reservoir sometimes act as a CO$_2$ sink, with fluxes dropping as low as -240.21 mmol.m$^{-2}$ d$^{-1}$ in 2019. The average CO$_2$ fluxes were lower than during the 2009-2011 period
(Table 1).

In 2009, the nighttime CO$_2$ emissions were significantly higher than daytime emissions, owing to a number of high fluxes reaching up to 493 mmol m$^{-2}$ d$^{-1}$ during the night. On average, nighttime emissions were recorded at 162 mmol m$^{-2}$ d$^{-1}$, surpassing the daytime average of 114 mmol m$^{-2}$ d$^{-1}$. In contrast, the diurnal variation in CO$_2$ fluxes was less pronounced in the following years. In 2010, there was minimal difference between daytime and nighttime fluxes, with average values of 170
and 185 mmol m$^{-2}$ d$^{-1}$, respectively. Similarly, in 2011, the variation was insignificant, with daytime fluxes averaging 77 mmol m$^{-2}$ d$^{-1}$ and nighttime fluxes at 70 mmol m$^{-2}$ d$^{-1}$.

After upscaling to the entire reservoir area (Figure 3A) and on a monthly basis, two out of three campaigns in the first period indicated that CH$_4$ emission from EC measurements were higher than those calculated directly from the two terms: ebullition and diffusion. This was the case in May 2009 (EC: 1.51 ± 0.12 Gg CH$_4$ month$^{-1}$, GE: 0.73 ± 0.11 Gg CH$_4$ month$^{-1}$) and March
2011 (EC: 1.31 ± 0.10 Gg CH$_4$ month$^{-1}$, GE: 1.25 ± 0.23 Gg CH$_4$ month$^{-1}$). This the opposite during the March 2010 campaign with a ~40% higher emission from GE (1.81 ± 0.59 Gg CH$_4$ month$^{-1}$) than from EC extrapolated measurement (1.31 ± 0.10 Gg CH$_4$ month$^{-1}$). Similarly, GE calculated emissions were higher than the EC extrapolated ones in 2019 (EC: 0.34 ± 0.07 Gg CH$_4$ month$^{-1}$, GE: 0.66 ± 0.03 Gg CH$_4$ month$^{-1}$) and 2022 (EC: 0.44 ± 0.05 Gg CH$_4$ month$^{-1}$, GC: 1.39 ± 0.05 Gg CH$_4$ month$^{-1}$).




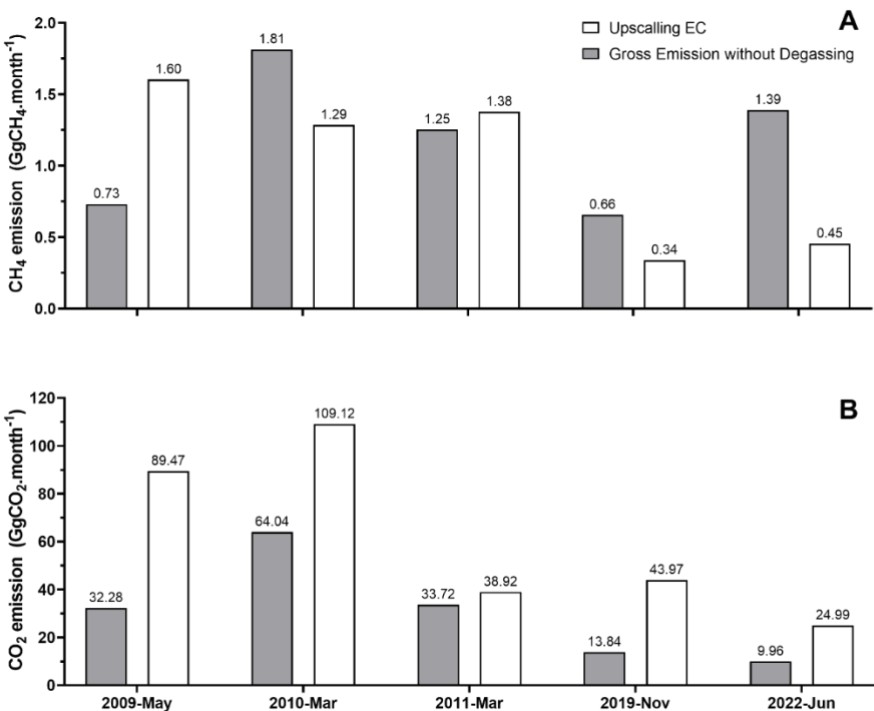

**Figure 3: Comparisons of methane (A) and carbon dioxide (B) emissions by upscaling eddy covariance method (white) and gross emissions from the reservoir's water surface without degassing (grey) from five field campaigns in Nam Theun 2 Reservoir**

For $CO_2$ emissions, upscaled EC measurements consistently showed higher values than GE calculations (Figure 3B). The

trends between the two methods remained similar throughout the sampling periods. Specifically, in 2009 and 2019, EC estimations were roughly three times higher than GE calculations. In 2012 and 2022, the EC values were approximately twice and 2.5 times higher, respectively. Notably, in 2011, the difference was significantly smaller, with EC measurements only being about 10% higher than GE calculations (Figure 3B).

### 3.2 Temporal variations and vertical profiles of temperature, dissolved oxygen, greenhouse gas concentrations and
**carbon species in the reservoir**

A set of vertical profiles from RES4 (Figure 4) was selected to represent the temporal dynamics of the reservoir water column over time. This station was the most comprehensively sampled for the study period, and it was centrally located within the reservoir, making it the best representative of overall trends. Deshmukh et al. (2014; 2016; 2018) and Guérin et al. (2016) have thoroughly described data up to 2012, therefore, we will focus on data from 2013 to 2022 (Fig. 4). Data are expressed in the

format of Mean ± SD.

Over the 14-year monitoring, the reservoir water column was a thermally stratified (Fig. 4) during the warm seasons (from mid-February to mid-October). This stratification was particularly strong during the WD season, creating a sharp thermocline at a depth of 4.23 ± 1.62 m, with temperature differences between the epilimnion and hypolimnion averaging 7.51 ± 3.51 °C



(27.53 ± 2.56 °C and 20.02 ± 2.40 °C, respectively; p<0.05). The stratification intensity diminished during the WW season,

resulting in a deeper thermocline at 5.00 ± 2.75 m with a temperature difference of 5.57 ± 2.76 °C between surface and bottom

areas (27.69 ± 1.86 °C and 22.12 ± 2.04 °C, respectively; p<0.05). During the CD season, the reservoir experienced an overturn,

leading to a well-mixed water column with an average temperature of 21.24 ± 2.28 °C (p<0.05) all through the water column,

as described in Chanudet et al., 2012 and Guérin et al., 2016.

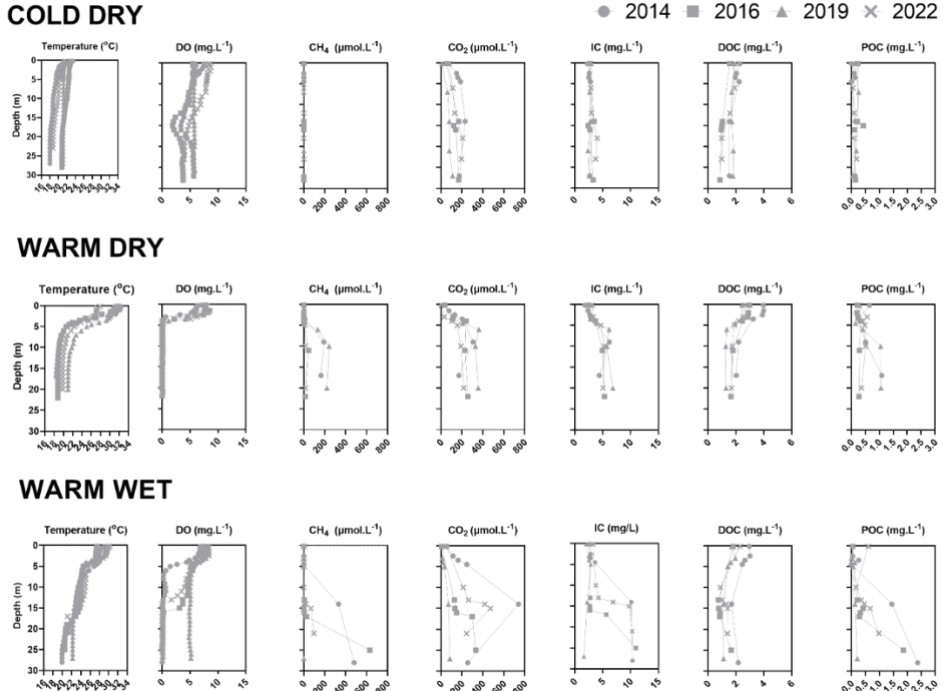

**Figure 4: Vertical profiles of temperature (°C), dissolved oxygen (mg L$^{-1}$), methane (µmol L$^{-1}$), carbon dioxide (µmol L$^{-1}$), inorganic carbon (mg L$^{-1}$), dissolved organic carbon (mg L$^{-1}$) and particulate organic carbon (mg L$^{-1}$) at station RES4. The figure shows data from representative dates in the years 2014 (circle), 2016 (square), 2019 (triangle) and 2022 (cross).**

When the water column was thermally stratified, an oxycline established at approximately the same depth as the thermocline

(Fig. 4). However, oxygen could accumulate in a thicker layer during the WW season. Both dry and wet warm seasons featured

an oxygen-rich epilimnion, with average O$_2$ concentration of 6.83 ± 1.21 mg L$^{-1}$ (86.97 ± 15.96 % saturation) (p<0.05) in the

WD season and 6.71 ± 1.30 mg L$^{-1}$ (85.76 ± 17.59% saturation) (p<0.05) in the WW season. In contrast, the hypolimnion was

anoxic below the oxycline. Sporadically, during the WW season, the water column mixed and, as a consequence, the bottom

was oxygenated, as observed in 2019 at RES4 station (Fig. 4). During the CD season, the entire water column was well-

oxygenated, with oxygen concentration decreasing gradually from the surface (6.73 ± 1.71 mg L$^{-1}$ - 78.41 ± 19.86 % saturation;

p<0.05) to the bottom (3.09 ± 2.09 mg L$^{-1}$ - 33.73 ± 31.48 % saturation; p<0.05). These seasonal trends remained consistent

throughout the studied period.



Methane concentrations (Fig. 4) in the water column ranged from 0.0025 to 1325.98 µmol L$^{-1}$ during the observation period. During the warm seasons, when thermal stratification occurred, concentrations of CH$_4$ increased significantly below the thermocline in suboxic-anoxic waters. Bottom concentrations were on average 145.92 ± 183.93 µmol L$^{-1}$ (p<0.05) and 228.86

± 252.56 µmol L$^{-1}$ (p<0.05) respectively during the WD and WW (p<0.05), for surface concentrations respectively of 6.98 ± 27.89 µmol L$^{-1}$ (p<0.05) and 3.01 ± 16.47 µmol L$^{-1}$ (p<0.05). In the CD season, a strong decrease in the CH$_4$ concentration in the hypolimnion, (down to 99.94 ± 212.86 µmol L$^{-1}$; p<0.05) and in the epilimnion (down to 2.21 ± 10.48 µmol L$^{-1}$; p<0.05) was observed. As a result, the CH$_4$ concentrations in the water column were the lowest in the cold season (37.41 ± 124.66 µmol L$^{-1}$, p<0.05) compared to the warm seasons (71.54 ± 126.48 µmol L$^{-1}$; p<0.05 in WD and 77.32 ± 163.18 µmol L$^{-1}$;

p<0.05 in WW). Average concentrations of CH$_4$ in the water column peaked before the reservoir commissioning in 2009, at 156.05 ± 193.27 µmol L$^{-1}$ (surface: 3.43 ± 15.02 µmol L$^{-1}$ – bottom: 295.41 ± 241.66 µmol L$^{-1}$). After the commissioning in April 2010, average concentrations in the water column declined to 125.52 ± 195.73 µmol L$^{-1}$ (surface: 9.3 ± 31.34 µmol L$^{-1}$ – bottom: 251.75 ± 286.17 µmol L$^{-1}$), and were three time lower in the following year 2011 (42.21 ± 92.86 µmol L$^{-1}$). This decreasing trend continued for the duration of the monitoring, hitting the lowest points in 2021 at 7.03 ± 18.23 µmol L$^{-1}$

(surface: 0.55 ± 2.24 µmol L$^{-1}$ – bottom: 17.50 ± 28.38 µmol L$^{-1}$), and in 2022 at 9.32 ± 27.44 µmol L$^{-1}$ (surface: 0.29 ± 1.07 µmol L$^{-1}$ – bottom: 29.03 ± 47.47 µmol L$^{-1}$).

Over the 14 years of measurement, carbon dioxide concentrations (Fig. 4) fluctuated between 0.11 and 2775.38 µmol L$^{-1}$ across all depths of the reservoir, averaging 183.10 ± 172.13 µmol L$^{-1}$ (p<0.05) in the WD, 212.34 ± 194.95 µmol L$^{-1}$ (p<0.05) in the WW and 167.50 ± 159.53 µmol L$^{-1}$ (p<0.05) in the CD. Unlike CH$_4$, CO$_2$ surface concentrations increased from the warm

seasons (86.11 ± 88.72 µmol L$^{-1}$ in WD and 97.33 ± 81.25 µmol L$^{-1}$ in the WW; p<0.05) to the cold season (100.26 ± 83.28 µmol L$^{-1}$; p<0.05) when the overturn occurred. During the warm seasons, in the presence of thermal stratification, CO$_2$ concentrations rose significantly below the thermocline. In the hypolimnion, CO$_2$ showed a similar build-up to CH$_4$ in WD (242.94 ± 198.40 µmol L$^{-1}$) and WW seasons (309.76 ± 274.20 µmol L$^{-1}$) before dropping in CD seasons (224.19 ± 237.65 µmol L$^{-1}$) (p<0.05). CO$_2$ levels in the water column were the highest in the first fully-operated year 2010, at 304.62 ± 237.65

µmol L$^{-1}$ (surface: 135.23 ± 97.73 µmol L$^{-1}$ - bottom: 451.94 ± 400.73 µmol L$^{-1}$), and then decreased annually, reaching the lowest values in 2021 at 85.39 ± 91.28 µmol L$^{-1}$ (surface: 37.89 ± 44.43 µmol L$^{-1}$ – bottom: 107.41 ± 83.80 µmol L$^{-1}$). As opposed to CH$_4$ which follow a continuous decrease all along the observation period, an unexpected increase of CO$_2$ was observed in 2022 (+45% or 124.33 ± 102.30 µmol L$^{-1}$ - surface: 71.52 ± 52.04 µmol L$^{-1}$ - bottom: 142.70 ± 84.15 µmol L$^{-1}$).

IC concentrations (0.50 – 54.96 mg L$^{-1}$, Fig. 4) showed an increase trend from the surface to the bottom across all three seasons.

The differences between two layers were more substantial during the warmer seasons with the presence of a thermocline (surface: 3.54 ± 1.30 mg L$^{-1}$ in WD, 2.90 ± 1.38 mg L$^{-1}$ in WW; bottom: 6.23 ± 3.56 mg L$^{-1}$ in WD, 6.79 ± 5.64 mg L$^{-1}$ in WW; p<0.05) compared to the CD season (surface: 2.88 ± 1.06 mg L$^{-1}$ – bottom: 4.82 ± 4.67 mg L$^{-1}$; p<0.05). Thus, the average amount of IC in the water column decreased from WD (4.88 ± 2.88 mg L$^{-1}$) to WW (4.34 ± 3.58 mg L$^{-1}$) to CD (3.69 ± 2.73 mg L$^{-1}$) (p<0.05). Over time, IC concentration in the water column peaked in 2010, at 6.23 ± 4.74 mg L$^{-1}$, and declined

to the lowest point in 2018, at 2.98 ± 1.56 mg L$^{-1}$ and gradually rose again to 3.95 ± 1.51 mg L$^{-1}$ in 2022.





DOC concentrations (0.50 – 8.87 mg $L^{-1}$, Fig. 4) presented a contrasting trend between the epilimnion and the hypolimnion. DOC levels were higher in the epilimnion across all three seasons (WD: 2.90 ± 1.02 mg $L^{-1}$, WW: 2.36 ± 0.93 mg $L^{-1}$, and CD: 1.99 ± 0.66 mg $L^{-1}$; p<0.05) compared to the hypolimnion (WD: 2.05 ± 0.68 mg $L^{-1}$, WW: 1.90 ± 0.67 mg $L^{-1}$, CD: 1.64 ± 0.69 mg $L^{-1}$; p<0.05). The differences of concentrations between the hypolimnion and the epilimnion in the water column

increased with the intensity of the thermocline (WD>WW>CD). The average DOC concentration in the water column followed the same seasonality with the highest concentration at 2.53 ± 0.98 mg $L^{-1}$ in the WD season, followed by WW at 2.05 ± 0.82 mg $L^{-1}$, and the lowest during the CD season at 1.81 ± 0.68 mg $L^{-1}$ (p<0.05). Interannually, similar to IC, DOC concentrations peaked in 2010 (2.47 ± 0.99 mg $L^{-1}$) in the water column, gradually decreased to hit the lowest in 2018 (1.35 ± 0.49 mg $L^{-1}$) and then, increased again to 1.88 ± 0.61 mg $L^{-1}$ in 2021 and 1.89 ± 0.42 mg $L^{-1}$ in 2022.

POC concentrations (0.01 – 14.31 mg $L^{-1}$, Fig. 4) in the surface layer (WD: 0.31 ± 0.50 mg $L^{-1}$, WW: 0.24 ± 0.34 mg $L^{-1}$ and CD: 0.16 ± 0.24 mg $L^{-1}$; p<0.05) was consistently lower than in the bottom layer in all three seasons (WD: 0.68 ± 0.78 mg $L^{-1}$, WW: 1.12 ± 1.23 mg $L^{-1}$ and CD: 0.56 ± 1.03 mg $L^{-1}$; p<0.05). The warm seasons had higher POC levels in the water column compared to the cold season (WD: 0.44 ± 0.60 mg $L^{-1}$, WW: 0.47 ± 0.78 mg $L^{-1}$ and CD: 0.30 ± 0.63 mg $L^{-1}$) (p<0.05). Concentrations were 0.53 ± 0.67 mg $L^{-1}$ in 2010, peaked in 2012 at 0.60 ± 0.89 mg $L^{-1}$, then declined to 0.11 ± 0.16 mg $L^{-1}$ in

2017. Afterward, POC concentration rose, nearly matching the 2012 peak, at 0.59 ± 1.16 mg $L^{-1}$ in 2022. During these series of changes, the epilimnion and the hypolimnion showed opposite trends, i.e., whereas the POC concentration in the surface layer increased from 0.29 ± 0.40 mg $L^{-1}$ in 2010 to 0.51 ± 1.01 mg $L^{-1}$ in 2022, bottom layer POC decreased from the peak of 1.18 ± 1.33 mg $L^{-1}$ in 2012 to 0.64 ± 1.20 mg $L^{-1}$ in 2022. In both layers, 2017 had the lowest POC concentrations with an average 0.11 mg $L^{-1}$.

**3.3 Seasonal and temporal variations of $CH_4$ and $CO_2$ emissions from the reservoir**

Over the 14-year monitoring period (Fig. 5A), the reservoir water released a total of 202.5 Gg $CH_4$ or around 5468 Gg $CO_2$eq. The amount of $CH_4$ released from the reservoir water surface decreased over time after peaking at 585.4 Gg $CO_2$eq year$^{-1}$ in 2010. By the end of the monitoring period, $CH_4$ emission were halved in 2021 (282.1 Gg $CO_2$eq) (Fig. 6). Most of this reduction resulted from decreases in diffusion, which dropped by 97.5% in 2022, compared to their peaks at the beginning of the study

period (2009-2010) (Fig. 5A). In contrast, $CH_4$ ebullition did not exhibit significant variation (p = 0.77, ranging from 227.6 to 354.9 Gg $CO_2$eq year$^{-1}$) and remained the dominant pathway each year since 2010. This was especially notable after 2020, when bubbling made up more than 94 to 97.5% of the total $CH_4$ emissions (Fig. 5A).

Among the three seasons (Fig. 5A), $CH_4$ emissions were the highest during the WD season (approximately 2582 Gg $CO_2$eq – 47%), while the WW (approximately 1324 Gg $CO_2$eq – 24%) and the CD (approximately 1562 Gg $CO_2$eq – 29%) seasons

showed no significant difference in $CH_4$ emissions when compared together (p = 0.12). The primary pathway for $CH_4$ emission was ebullition, which contribute to 76.8% of the total emissions. This pathway was most significant during the WD season, when approximately 50% of ebullition happened, while the WW and CD seasons shared the remaining portion equally. In the





course of the 14-year study, diffusion from the reservoir water surface took up 18.6% of the total $CH_4$ emission. The CD season emitted the most through diffusive fluxes with 468.2 Gg $CO_2$eq (46%), followed by a decrease from WD (318.8 Gg $CO_2$eq –

31%) to WW (228.0 Gg $CO_2$eq – 23%). Degassing contributed the least to $CH_4$ emissions, accounting for only 4.6% with approximately 98% of this amount occurring during the warm seasons. From 2009 to 2022, $CH_4$ emissions via degassing and ebullition during the warm seasons did not exhibit statistically significant variation (degassing: $p = 0.10$ for WD and $p = 0.11$ for WW; ebullition: $p = 0.24$ for WD and $p = 0.33$ for WW). In contrast, diffusive fluxes during both warm seasons showed significant variation ($p<0.05$). During the CD season, all three emission pathways, diffusion, degassing, and ebullition, varied

significantly over the study period. Consequently, total $CH_4$ emissions during the WW season did not change significantly over time ($p = 0.13$), whereas significant temporal variation was observed in $CH_4$ emissions during the WD and CD seasons ($p<0.05$ for both).

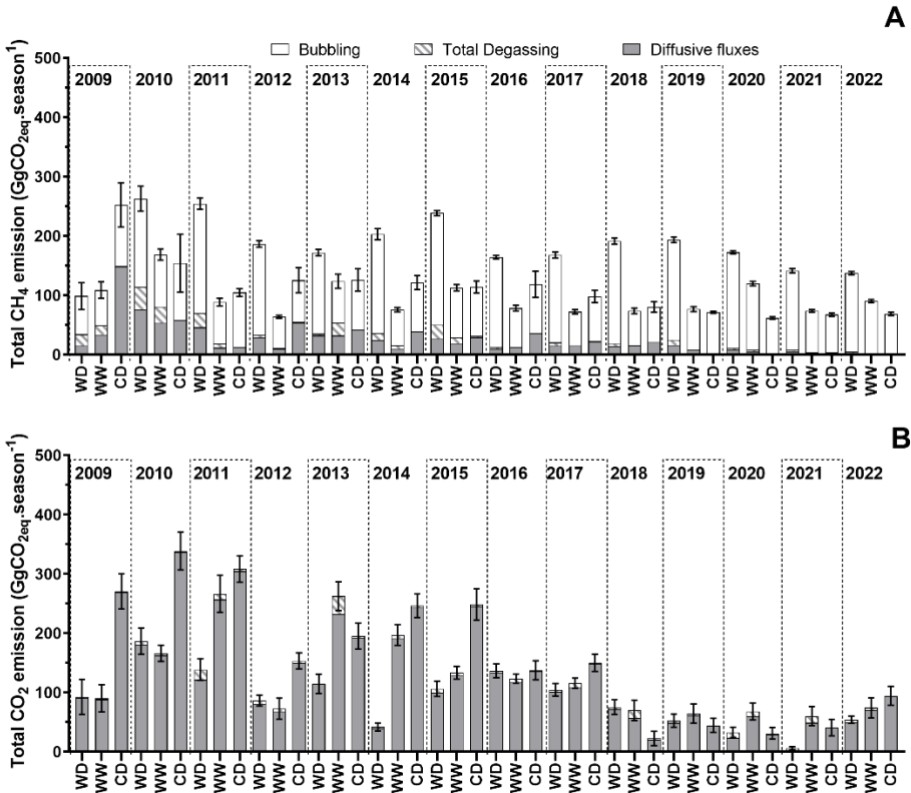

**Figure 5: Seasonal variations from 2009 to 2022 of methane (A) and carbon dioxide (B) emissions in Gg of $CO_2$ equivalent by season**
**from the three main pathways assessed: diffusive fluxes (grey), degassing (hatched) and ebullition (white). The error bar represents the total SE from all pathways combined.**

Over the course of study, the total $CO_2$ emission (Fig. 5B) amounted to approximately 5268 Gg $CO_2$eq. The decrease in $CO_2$ emission by 87% from 2011 (712.9 Gg $CO_2$eq year$^{-1}$) to 2021 (106.5 Gg$CO_2$eq year$^{-1}$) (Fig. 6) was primarily driven by diffusion at the surface of the reservoir. Degassing of $CO_2$ (209 Gg $CO_2$eq in 14 years) did not show significant differences from 2009





to 2022 (p = 0.232, ranging from 4.56 to 33.05 Gg $CO_2$eq year$^{-1}$). Annually, $CO_2$ emissions from the reservoir gradually decreased (p<0.05), with the exceptions of 2012 and 2018, when emissions suddenly dropped by half compared to the previous years. The year 2022 was also notably different with a doubling of the emissions that year compared to the ones calculated for 2021.

Among the three seasons, approximately 43% were released during the CD season, 23% during the WD season and the

remaining 34% during the WW season (Fig. 5B). These trends mirrored the $CO_2$ emission by diffusion at the air-water interface which was the dominant pathway, accounting for 96% of the total emission of $CO_2$. As for $CH_4$, with approximately 4% of the total $CO_2$ emission, degassing was a minor pathway, around 87% being released occurring during the warm seasons.

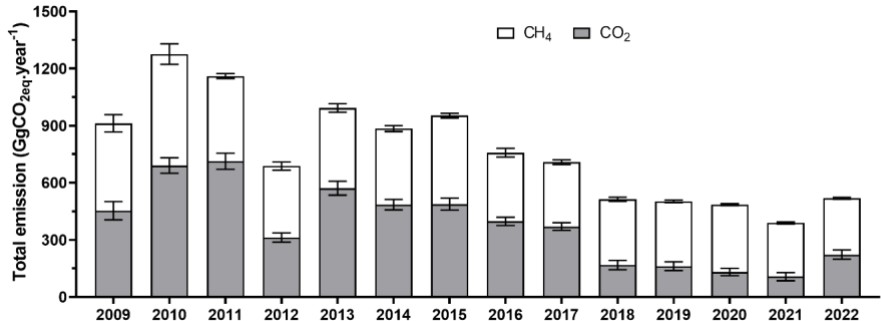

**Figure 6: Annual evolution from 2009 to 2022 of the total greenhouse gases ($CH_4$ in white and $CO_2$ in grey) emission in Gg of $CO_2$**
**equivalent by year. The error bar represents the SE of each gas.**

The total gross emissions of both $CO_2$ and $CH_4$ together over the study period (2009-2022) were 10736 Gg $CO_2$eq, with an approximately equal distribution from both gases ($CO_2$: 5268 Gg $CO_2$eq – 49%; $CH_4$: 5468 Gg $CO_2$eq – 51%). Emissions were the highest in 2010, at 1276 Gg $CO_2$eq year$^{-1}$ and decreased over time to the lowest point in 2021, at 389 Gg $CO_2$eq year$^{-1}$ (Fig. 6). In 2009, $CH_4$ and $CO_2$ contributed equally to gross emissions. From 2010 to 2017, $CO_2$ was the dominant emission source

accounting for approximately 50–60% of total emissions largely due to dominant diffusive fluxes (Fig. 5B); whereas $CH_4$ became the primary contributor from 2018 to 2022 due to constant and high ebullition rates as the main emission pathway (Fig. 5A).

Seasonal analysis showed that the highest gross emissions occurred during the CD season, totalling 3841 Gg $CO_2$eq, followed by the WD season with 3809 Gg $CO_2$eq, and the WW season with 3086 Gg $CO_2$eq. $CO_2$ was the dominant greenhouse gas

during the WW and CD seasons, contributing 57% and 59% of total seasonal emissions, respectively, with diffusion at the air-water interface serving as the dominant pathway. In contrast, $CH_4$ was the major contributor during the WD season, making 68% of the seasonal emission, particularly through ebullition, which remained the primary pathway throughout the entire study period. Degassing represented only 4.3% of total $CO_2$eq emissions (considering both $CO_2$ and $CH_4$), although its contribution was more pronounced during warm periods.





## 4 Discussions

### 4.1 Diurnal variation of CH$_4$ and CO$_2$ fluxes from EC measurement

The CH$_4$ diurnal variations from the first three campaigns were discussed in the study of Deshmukh et al. (2014), in which they indicated a strong relationship between hydrostatic pressure and methane emissions, particularly through ebullition, at the NT2 reservoir.

An unique bimodal diurnal variation of CH$_4$ emissions was attributed mainly to the changes in atmospheric pressure, which were linked to the semidiurnal pattern of atmospheric pressure influenced by global atmospheric tides (Deshmukh et al., 2014). Specifically, CH$_4$ emissions peaked around midnight and again near midday, following periods of atmospheric pressure drop, a phenomenon shown to trigger ebullition by reducing hydrostatic pressure at the reservoir bottom (Deshmukh et al., 2014). This effect was significant at lower water depths, where a greater relative change in total static pressure further facilitates gas release (Deshmukh et al., 2014).

Additionally, this CH$_4$ diurnal variation, which daytime fluxes were higher than nighttime fluxes was found by Bastviken et al. (2010) and Tan et al. (2021). The latter finding claimed that solar radiation played an important role in the process of aerobic methanogenesis, also known as "methane paradox" in which dissolved CH$_4$ concentration is supersaturated in oxic water (Tang et al., 2016). Even though, the production of methane in oxygen-rich waters is not yet fully understood, recent researches indicate a strong association with algal photosynthesis, where algal growth drives CH$_4$ production (Bogard et al., 2014; Hartmann et al., 2020). Bižić et al. (2020) recent conducted experiments using light and dark conditions, which suggested that cyanobacteria contribute notably to CH$_4$ production, directly linking this process to light-dependent primary productivity. As a matter of consequence, light availability, a major factor in photosynthetic activity (Krause & Weis, 1991), indirectly influences CH$_4$ production, while temperature plays a critical role in the organic decomposition associated with methanogenesis (Yvon-Durocher et al., 2014). Other studies in wetland environments have shown that primary productivity significantly modulate the daily CH$_4$ emission cycle with increased fluxes during the day (Hatala et al., 2012; Mitra et al., 2020). Furthermore, solar radiation appears to suppress CH$_4$ oxidation in the epilimnion, increasing net CH$_4$ emissions by limiting its breakdown in sunlit surface waters (Murase & Sugimoto, 2005; Tang et al., 2014).

Nighttime high CO$_2$ emissions during low wind speed conditions (Supplement Fig. S4 in Deshmukh et al., 2014) might result from waterside convection and erosion of the thermocline (Eugster et al., 2003; Jeffery et al., 2007; MacIntyre et al., 2010; Mammarella et al., 2015; Podgrajsek et al., 2015). Daytime lower fluxes could also result from the photosynthesis activity in the euphotic water column, therefore leading to a decrease of CO$_2$ concentrations in the upper water column.

During the daytime, absorption of solar radiation in the water column diminishes the depth of the surface mixing layer and thereby thermal convection from heat loss (Eugster et al., 2003; Podgrajsek et al., 2015). CO$_2$ fluxes were lower during the 2011 field campaign, even though it exhibited wind speed almost as high as in 2010 (up to 8 m s$^{-1}$). With low surface water temperature and with a weak thermal gradient, it seems than even high wind conditions during the daytime were not able to overcome with the heating effect.





The diurnal variation in $CO_2$ emissions can also be explained by temperature-driven increases in respiration, decomposition and photooxidation of organic matter and DOC in the water column. Respiration increases $CO_2$ concentrations in the surface

layer, leading to higher emissions, especially at night when photosynthetic $CO_2$ uptake from phytoplankton stops, allowing $CO_2$ to accumulate and be released. This balance between $CO_2$ consumption via photosynthesis (daytime) and $CO_2$ production via respiration (nighttime and daytime) (Spank et al., 2023) is in agreement with the observation in this study, especially during 2009 in which significantly higher nighttime $CO_2$ emissions were observed, when photosynthesis likely played a significant role in reducing daytime $CO_2$ fluxes.

Particularly during a strong thermal stratification period, sampling only during daytime could lead to a bias underestimating the daily flux. Exclusion of the diurnal variability in $CO_2$ fluxes, for instance not sampling nighttime fluxes, during a strong thermal stratification period could underestimate the annual mean flux by 3.5%. This percentage applies for the NT2 reservoir knowing that it is thermally stratified for the three months of the warm dry season (Chanudet et al., 2016; Guérin et al., 2016). However, in this study, no correction was applied to account for the potential enhancement of nighttime fluxes. As a result,

while our estimates were considered robust, they should be interpreted as conservative due to the uncorrected diurnal variability.

High wind events captured in March 2010 (Supplement Fig. S4 in Deshmukh et al., 2014) were 70% higher than average wind speed and triggered very high $CO_2$ fluxes, up to 616 mmol m$^{-2}$ day$^{-1}$. Under these circumstances, $CO_2$-rich hypolimnetic waters were likely to be brought up to the surface water, leading to substantially larger $CO_2$ fluxes, as compared to calm wind period.

Generally, diurnal EC measurements revealed that both $CH_4$ and $CO_4$ fluxes were underestimated if nighttime emissions were excluded. This effect was important under strong stratification during the WD season. Continuous high-frequency monitoring was therefore necessary to capture true daily flux dynamics.

### 4.2 Evaluating the differences between EC and discrete sampling upscaling (GE)

The position of the EC tower in the first three campaigns (Section 2.2) represents an ideal situation for measurement technique

comparison, particularly during mixed water column conditions. However, fluxes measured with EC on one side, and GE measurements (TBL and bubbling) methods on the other side are representative different spatial scales of emission sources. Furthermore, since EC fluxes are time-and-space integrated, one cannot anticipate 1:1 comparison with GE measurements (Erkkilä et al., 2018).

According to Spank et al. (2023), inland water emissions $CO_2$ and $CH_4$ are commonly measured using methods such as manual

gas sampling, floating chambers, and bubble traps (i.e. submerged funnels for this study). While these techniques deliver precise measurements, they often lack broader spatial and temporal coverage. In our case, the comprehensive sampling strategies and a set of sampling sites covering the variety of flooded ecosystems with a specific attribution of area to special sites (Section 2.2) accounted for the special variation. Alternatively, the EC approach enables continuous monitoring of gas fluxes across a larger footprint (up to 500 m; Deshmukh et al., 2014) and allow a better representativeness of the measured

emissions and the exchange processes (Spank et al., 2023).




For $CH_4$, in 2009, substantial rainfall during EC measurements (78.5mm during the EC deployment, i.e. 67% of total 2009 WD precipitation, Fig. 2E), likely influenced $CH_4$ emissions by enhancing surface water mixing. This could have increased diffusion that EC effectively captured ($1.51 \pm 0.12$ Gg $CH_4$ month$^{-1}$). In contrast, GE ($0.73 \pm 0.11$ Gg $CH_4$ month$^{-1}$) in May - 2009 were based on data with a limited spatial resolution for both water samplings and bubble traps. Therefore, calculations

relied mostly on gap-filling and ANN which might have introduced bias and led to underestimations in GE. By March 2010, the reservoir was commissioned, creating a $CH_4$ emission hotspot at RES9 (Guérin et al., 2016), due to increased turbulence, which kept the 3 km² area in constant mixing. This mixing brought $CH_4$-rich bottom water to the surface and triggered high diffusive fluxes. RES9 contributed 31.7% of total $CH_4$ diffusive fluxes in 2010 (Guérin et al., 2016) which is the highest proportion in the 14-year period, 100 times higher than in 2009 (when water was not sent to turbine, before commissioning).

2010 also was the year with highest $CH_4$ diffusive fluxes in the course of study. As the EC was located remotely from RES9 station, the EC ($1.31 \pm 0.10$ Gg $CH_4$ month$^{-1}$) was unable to capture the hotspot of $CH_4$ diffusion. The EC estimation of the $CH_4$ emissions from the reservoir was necessarily lower than the GE ($1.81 \pm 0.59$ Gg $CH_4$ month$^{-1}$) approach. In 2011, the reduced diurnal difference in the EC fluxes (1.3 times higher during the day) might explain why EC ($1.31 \pm 0.10$ Gg $CH_4$ month$^{-1}$) and GE ($1.25 \pm 0.23$ Gg $CH_4$ month$^{-1}$) results were more comparable, as the water sampling for GE calculations

occurred only during daytime (10:00 to 16:00). The latter two campaigns saw a relocation of the EC station. The tower was placed onshore, which limited its ability to capture both diffusive and ebullitive fluxes as these were often filtered due to unsuitable wind directions. Deemer et al. (2016) indicated that $CH_4$ fluxes measured as a combination of bubbling and diffusion were approximately twice as high as those from diffusion alone, consistent with the difference observed between EC and GE measurements in these years (Sector 3.3). An important consideration in interpreting the results is the role of $CH_4$ bubbling,

which became increasingly significant during the latter half of the study period (Fig. 5). While the EC station captured fluxes continuously over time, it was limited to a single point and depth. In contrast, the discrete chamber measurements, conducted across various depths and reservoir locations, offered broader spatial coverage. Notably, because bubbling was measured over a full day, its temporal resolution was comparable to that of the EC method. Therefore, the key distinction lies in spatial variability: chamber measurements captured a more representative site of bubbling across the reservoir, which was crucial for

understanding the spatial heterogeneity of emissions.

$CO_2$ fluxes measured by the EC method consistently exceeded those calculated using the TBL model (GE). This difference, also reported in previous studies (Erkkilä et al., 2018; Mammarella et al., 2015), rooted from differences in spatial resolution and sampling location between the two methods, which captured spatial variability in $CO_2$ fluxes across the lake differently (Scholz et al., 2021). The EC station, positioned near (average depth: 10 m in 2009, 10.5 m in 2010 and 10.5 m in 2011,

Deshmukh et al., 2014) or on (2019, 2022) the reservoir shore, primarily captured fluxes from shallow water areas. On the opposite, water samples used in TBL-based diffusive flux calculations were gathered in deeper sites of the reservoir. This spatial variability is consistent with findings from other research showing that $CO_2$ fluxes are generally higher in shallow regions (Loken et al., 2019; Xiao et al., 2020). Shallow areas often experience warmer water temperatures, which can stimulate microbial respiration and elevate $CO_2$ emissions. Additionally, in shallow zones, sediments lie closer to the water surface, and



the mixing layer more readily extends to the lake bottom (Holgerson, 2015). Organic material inputs from the surrounding land, providing a rich substrate for microbial respiration and subsequent $CO_2$ production can be deposited close to the shoreline (Xiao et al., 2020). Thus, these spatial and environmental differences between EC and discrete sampling sites likely contributed to the higher $CO_2$ fluxes observed in the EC upscaling method. Additionally, for the earlier campaigns (2009-2011), the consistently higher EC fluxes likely result from methodological differences, including the EC method ability to capture real-

time turbulent fluxes, short-term peak events (for instance, wind bursts or convective mixing), and continuous diurnal variation, in contrast to the TBL method reliance on daytime sampling and calculated gas transfer velocities. Potential underestimation of the gas transfer coefficient $k_T$ in TBL calculations, especially under calm conditions or in the absence of direct turbulence measurements, could lead to systematically lower flux estimates. Thus, while spatial heterogeneity explains some of the differences in later years, methodological limitations inherent to the TBL approach likely played a central role in earlier

campaigns.

In short, the results highlighted the importance of integrating multiple measurement approaches to reduce spatial and temporal biases in GHG emission estimates. By taking advantages of the continuous temporal resolution offered by EC and the broader spatial coverage achieved through discrete sampling, the overall accuracy and representativeness of emission assessments could be enhanced. Nonetheless, in this study, due to the limited number of EC campaigns and the extensive dataset available

from discrete sampling, the calculation of total GHG emissions from the reservoir water relied primarily on discrete sampling data.

### 4.3 Temporal dynamics of $CH_4$ emissions from the reservoir water surface

The seasonal variations in $CH_4$ emissions from the NT2 reservoir illustrate the critical role of thermal stratification, water mixing, and organic matter decomposition in regulating greenhouse gas fluxes as described by (Guérin et al., 2016).

During the WD season, total emissions reached their maximum due to higher temperatures and lower dissolved oxygen levels in the water column, creating more favourable conditions for methane production (Upadhyay et al., 2023). Stratification during this period creates distinct layers within the water column, with limited vertical mixing. This vertical isolation enhances anaerobic decomposition of organic material, thereby fuelling methanogenesis in deeper layers. Ebullitive fluxes contributed predominantly in this season, consistent with the findings of Deshmukh et al. (2014). Ebullition is well-documented to be

positively correlated with temperature (Chanton et al., 1989; St. Louis et al., 2000; Zheng et al., 2022), as elevated temperatures promote $CH_4$ production and decrease $CH_4$ solubility. During the WD season, the decline in the water level reduces hydrostatic pressure on the sediment (Deshmukh et al., 2014; Maeck et al., 2014), triggering bubble release. Diffusive fluxes share similar driving parameters as ebullition, in which higher temperature and lower DO in the water column (Fig. 4) facilitate $CH_4$ to reach the surface as $CH_4$ oxidation is reduced (D'Ambrosio & Harrison, 2021). Moreover, the availability of organic matter

plays a crucial role; higher carbon content in the water column (DOC and IC) during the WD season due to processes such as enhanced microbial activities, increased discharges from watershed, stronger photosynthesis, contributes to $CH_4$ production by supplying labile allochthonous and autochthonous OM. Degassing of $CH_4$ was also highest during this period, consistent with the trend reported by Deshmukh et al. (2016) after 14 years of measurements. Degassing was significant only when





spillway release occurred during flooding in the later months of WD and early WW (Fig. 2D), when CH₄-rich water from
below the oxycline, approximately 15 m deep, was discharged (Deshmukh et al., 2016).

In contrast, methane emissions reached their lowest levels during the WW season, characterized by a twofold decrease in
ebullition and the lowest diffusive fluxes among the three studied seasons. The erosion of the thermocline during this period
promoted more aerobic conditions, which favoured methanotrophic activity and thereby reduced net CH₄ emissions.
Deshmukh et al. (2016) stated the reduced bubbling observed at the NT2 reservoir during the WW season to several interrelated
mechanisms: (1) CH₄ concentrations within bubbles were substantially lower than those recorded during the WD season, likely
due to increased oxidation of CH₄; (2) bubble CH₄ concentrations exhibited minimal variation with depth during the WW
season, unlike the WD season, where elevated concentrations were confined to shallower depths, indicating the role of depth-
dependent processes such as CH₄ dissolution, which can reduce CH₄ content in bubbles by up to 20% in waters less than 10
meters deep, as reported by McGinnis et al. (2006) and Ostrovsky et al. (2008); (3) lower temperatures during the WW season
(Fig. 2A) led to a decrease in the methanogenic activity and increased CH₄ solubility, thereby suppressing bubble formation
and release; and (4) higher water levels during WW increased hydrostatic pressure, further inhibiting ebullition. For diffusive
fluxes, elevated river inflow during this season (Chanudet et al., 2012) may have produced a dilution effect, further lowering
CH₄ concentrations in surface waters and reducing the efflux to the atmosphere. Increased precipitation likely also contributed
to slower organic matter decomposition by diminishing the thermal gradient between surface waters and the hypolimnion.
Sporadic mixing events enhanced oxygen penetration into deeper layers and flooded sediments, limiting CH₄ accumulation in
the hypolimnion (Guérin et al., 2016). The weakening of the thermocline during the WW season (Fig. 4) further facilitated the
downward transport of dissolved oxygen, reducing the amount of CH₄ reaching surface waters due to CH₄ oxidation. Degassing
was also reduced since bottom concentrations are lower (Guérin et al., 2016) and since the water discharge at the powerhouse
were reduced in order to initiate a water storage phase before WD and CD seasons (Fig. 2D).

During the CD season, vertical mixing occurred annually as surface water temperatures declined and approached those of the
bottom layers (Fig. 4). Simultaneously, the reservoir surface area remained consistently high, with minimal inflow and outflow
discharges (Fig. 2D). These conditions adversely affected CH₄ solubility and hindered the development of anoxic zones
required for methanogenesis (Deshmukh et al., 2014). Ebullition in the NT2 reservoir is negatively correlated with both water
depth and fluctuations in water level (Deshmukh et al., 2014), which accounts for the lower bubbling flux observed during the
CD season compared to the WD season. The relatively stable hydrostatic pressure (Schmid et al., 2017), in conjunction with
the large surface area and higher wind speeds (Fig. 2B) compared to the two warm seasons, increased the gas transfer velocity,
thereby enhancing diffusive CH₄ fluxes, which peaked during this season (Guérin et al., 2007). CH₄ diffusive fluxes during
this period were also significantly influenced by reservoir overturn, which occurs as thermal stratification collapses due to
cooling surface temperatures. Guérin et al. (2016) previously identified reservoir overturn in NT2 as a hot moment for CH₄
emissions, where deep, CH₄-rich waters mix with surface layers, intensifying methane transport and release. The breakdown
of the stratification promotes gas exchange between the hypolimnion and the atmosphere, resulting in elevated CH₄ diffusive
emissions. In contrast, degassing was negligible during this season, accounting for only about 0.4% of total seasonal emissions,





due to low concentrations, minimal water discharge, and limited flood events due to low precipitation (Fig. 2E). This observation aligns with the findings of Deshmukh et al. (2016) from the early post-impoundment years.

From 2009 to 2022, $CH_4$ emissions exhibited a gradual decline, with peak emissions recorded after commission (2010) dropping to nearly half by the end of the study (2022). Such trends underscore the influence of reservoir age on $CH_4$ emissions (Barros et al., 2011), as older reservoirs tend to emit less $CH_4$ due to the depletion of readily degradable organic matter with no major input beyond the initial allochthonous supply and limited autochthonous production.

Over a 14-year period, ebullition was the dominant pathway for $CH_4$ emission. This contribution is consistent with values
reported for inland waters, where ebullition represents approximately 62% to 84% of $CH_4$ emissions (Zheng et al., 2022), and for shallow lakes and ponds, where the contribution ranges from 50% to 90% (Attermeyer et al., 2016; Saunois et al., 2020). It also surpasses the average estimated contribution of 65% from reservoirs globally (Deemer et al., 2016). The persistence of ebullition as the primary emission pathway can be attributed to its dependence on the decomposition of the flooded organic matter and bubble formation. Although ebullition was anticipated to decline over time as the labile parts of flooded OM
decreased, observations revealed stable ebullition rates persisting since the initial four years following reservoir impoundment (Serça et al., 2016). This observation was further supported by the fact that, in the NT2 reservoir, the primary carbon pool resided within the sediment, similar to that observed at Petit Saut reservoir (Abril et al., 2005), consisting of $2.75 \pm 0.23$ Mt C of soil organic matter and $0.15 \pm 0.23$ Mt C of belowground biomass, out of a total flooded carbon stock of 5.12 Mt C (Descloux et al., 2011). From 2009 to 2022, cumulative $CH_4$ emissions via bubbling from the NT2 reservoir amounted to approximately
4200 Gg $CO_2$eq, equivalent to a stable emission rate of about 300 Gg $CO_2$eq year$^{-1}$ (approximately 6.6 Gg C year$^{-1}$, considering a GWP of 27 for $CH_4$). Assuming a constant emission rate, the cumulative carbon export through ebullition over 14 years is estimated at around 0.1 Mt C, which represented only a minor fraction of the initial carbon stock in the sediment. Therefore, the stability of ebullition rates over this study suggested that a substantial pool of less labile sedimentary OM would continue to sustain consistent $CH_4$ production.

$CH_4$ diffusion, the second most significant $CH_4$ emission pathway from the water surface (Serça et al., 2016), has declined steadily over time, likely due to gradual changes in reservoir dynamics, such as the diminishing availability of labile OM, a better oxygenation by higher concentration of DO and enhanced methane oxidation. The hypothesis of increasing $CH_4$ oxidation rate in later studied years could explain the increase of $CO_2$ concentrations in 2022, despite $CH_4$ concentrations in the water column continued to decrease. Given that the reservoir mixing regime remained relatively constant throughout the
14-year period, the observed reduction in diffusive emissions is likely driven by biological and chemical factors rather than shifts in physical mixing. This is supported by a documented decrease in both DOC and IC concentrations in the water column over the study period, indicating lower substrate availability for methanogenesis (Colas et al., 2020). A gradual reduction in methane production in deep waters would, in turn, result in a diminished supply of dissolved $CH_4$ available for diffusion.

Overall, $CH_4$ emissions peaked during the WD season, driven by stratification, low DO, and reduced/decreasing hydrostatic
pressure. Ebullition was the dominant and stable pathway across all years, while diffusion declined over time. This confirmed the persistent role of ebullition in long-term $CH_4$ emissions, even as the reservoir matured. This study, which quantified $CH_4$



ebullition over an unprecedented 14-year period, contributed to the understanding of long-term bubbling dynamics, a temporal scale rarely documented in existing reservoir emission studies.

**4.4 Temporal dynamics of $CO_2$ emission from the reservoir water surface**

$CO_2$ emissions from the NT2 reservoir showed distinct seasonal patterns influenced by temperature, water column stratification, and biological activity (Deshmukh et al., 2018). The vertical profiles of $CO_2$ in this study found to be consistent and similar in shape with previous finding of Deshmukh et al. (2018), and with measurements from other tropical reservoirs (Abril et al., 2005; Chanudet et al., 2011; Guérin et al., 2006). Since 96% of $CO_2$ emission from NT2 reservoir came from diffusive fluxes, the understanding of the variation of the dissolved $CO_2$ concentrations is crucial to interpret the seasonal and

interannual patterns of $CO_2$ emissions.

In the WD season, the shape of the vertical profiles of $CO_2$ concentration suggest that $CO_2$ is mostly produced at the bottom of the reservoir, in the OM pool of flooded soil and vegetation; and consumed by bacterial degradation, which is enhanced by high temperature (Fig. 2A) of the WD season (Abril et al., 2005; Barros et al., 2011; Chanudet et al., 2011; Deemer et al., 2016; Guérin et al., 2006; St. Louis et al., 2000). It is supported by the fact that the concentrations of Chlorophyll-A in the

NT2 reservoir (unpublished data) always peaked in the WD season, promoting consumption of $CO_2$ in the epilimnion. Additionally, diffusive fluxes correlated with reservoir area and wind speed as described with $CH_4$, therefore, the WD season with lowest wind speed and surface area (Fig. 2B, Fig. 2C) led to low fluxes.

The WW season exhibited intermediate emission levels, with sporadic mixing events (Guérin et al., 2016) contributing to periodic $CO_2$ fluxes to the atmosphere. The erosions of the thermocline as well as the oxycline during this season (Fig. 4) allow

DO to penetrate deeper into the deep water, where the pool of $CO_2$ production via enhanced aerobic respiration is located. The highest $CH_4$ oxidation rate recorded in the WW season (Guérin et al., 2016) also contributed to the gradual increase of $CO_2$ emissions.

During the CD season, $CO_2$ emissions peaked, with diffusion at the air-water interface as a major contributor. This elevated emission rate was primarily driven by overturn and complete mixing of the water column, which redistributed $CO_2$-rich waters

from the bottom layers to the surface, facilitating their release, contributed to the "hot moment" observed for $CH_4$ emissions (Guérin et al., 2016). Furthermore, complete mixing and oxygenation of the water column during the CD season supports more extensive aerobic decomposition of organic matter, pelagic respiration (Bastviken et al., 2004) and the oxidation of large amount of hypolimnion $CH_4$ going up the vertical water column, both of which contribute to enhance $CO_2$ production. Hydrologically, residence time is recognized as one of the most influential factors in river damming (Prairie et al., 2018),

significantly affecting the degradation rate of DOC (Dillon & Molot, 1997; Sobek et al., 2007). Carbon mineralization has been shown to positively correlate with residence time (Dillon & Molot, 1997). The CD season was consistently characterized by the longest residence times, due to minimal outflows and a larger volume (Fig. 2C, Fig. 2D). This coincided with the lowest DOC concentrations observed among the three seasons, suggesting enhanced carbon processing and mineralization during that period.





$CO_2$ degassing exhibited seasonal patterns similar to those observed for $CH_4$. Degassing was more pronounced during the warm seasons, driven by elevated $CO_2$ concentrations in the water column, high water discharges, and occasional use of the spillway. In contrast, during the CD season, when reservoir volume remains relatively stable, degassing was negligible, contributing only around 1% of seasonal CD emissions. This seasonal pattern has remained consistent over time. Deshmukh et al. (2016) similarly reported negligible $CH_4$ degassing during the CD season in the early years following reservoir

impoundment.

Over the course of the study, $CO_2$ emissions exhibited a pronounced and consistent decline. The reduction is primarily attributed to the decrease in diffusion emissions, which remained the dominant pathway for $CO_2$ emissions throughout the monitoring period. Such a pattern is characteristic of aging reservoirs (Barros et al., 2011) in tropical and subtropical regions, where emissions tend to decline over time due to a reduced rate of organic matter mineralization under both aerobic and

anaerobic conditions. As for $CH_4$, this reduction is driven by the depletion of labile organic matter originating from flooded soils and vegetation (Abril et al., 2005; Guérin et al., 2008).

The observed increase in $CO_2$ emissions toward the end of the study period in 2022 could be partially explained by abnormal emission measurements during 2021, particularly the sudden absence of emissions during the WD season and reduced emissions in the CD season (Fig. 5B). Specifically, the surface $CO_2$ measurements in the WD season of 2021 were significantly

impacted by disruptions related to the global pandemic, resulting in only 36 observations, compared to 49 in 2022 and 57 in 2020. Meanwhile, measurement efforts remained stable across the WW and CD seasons during the three-year period (WW: 50 in 2020, 53 in 2021, and 51 in 2022; CD: 51 in all three years). The reduced number of observations during the 2021 WD season likely compromised the accuracy of emission estimates, resulting in limited spatial coverage and potential seasonal biases. Additionally, enhanced $CH_4$ oxidation during the CD season of 2021 may also contribute to these trends. Notably, $CO_2$

emissions during the CD season in 2022 rose markedly to 94 Gg $CO_2$eq, more than doubling the value recorded in 2021 (41 Gg $CO_2$eq). Concurrently, $CH_4$ diffusive emissions during the CD season peaked in 2021 at 3.3 Gg $CO_2$eq, compared to 2.4 Gg $CO_2$eq in 2020 and 1.4 Gg $CO_2$eq in 2022, pointing to a possible shift in carbon gas dynamics during this interval. Despite extensive analyses, no clear hydrological (discharges) or meteorological (air temperature, relative humidity, solar radiation, windspeed, rainfall) factors were identified as explanatory variables for these changes. Consequently, the potential influence

of measurement errors could not be fully excluded and suggested further investigation.

In support of this, Deshmukh et al. (2018) reported that $CO_2$ effluxes from the NT2 reservoir were nine times greater than carbon inputs in 2010 and three times greater in 2013, indicating that internal carbon sources, primarily flooded organic matter (Deshmukh et al., 2014) and autochthonous production were the main contributors to $CO_2$ emissions, rather than allochthonous inputs. A similar pattern was also observed in the Petit Saut Reservoir (Abril et al., 2005; Guérin et al., 2008), further

reinforcing this mechanism of internal carbon-driven $CO_2$ production in tropical reservoirs.

In summary, $CO_2$ emissions declined steadily over 14 years, mainly due to reductions in diffusive fluxes as labile OM was depleted. The CD season remained the peak emission period due to water column overturn and respiration. This trend reflected the ageing process of the reservoir's carbon cycle.





**4.5 Temporal dynamics of gross emissions from the NT2 reservoir after 14 years of impoundment**

The observed temporal decline in total GHG emissions showed the progressive dynamic of biogeochemical processes as the reservoir matured. The initial post-impoundment peak corresponds to the rapid decomposition of labile OM inundated during the early flooding phase, a common trend in newly created reservoirs (Prairie et al., 2018; St. Louis et al., 2000).

The transition from $CO_2$ to $CH_4$ dominated emissions over time demonstrated a shift in the dominant pathways, from diffusive fluxes to ebullitive fluxes. This trend supports the concept of non-stationarity in GHG dynamics, where the relative importance

of pathways and gas species evolves with reservoir age.

The seasonal dynamics of gross GHG emissions were primarily governed by changes in the dominant gas species and emission pathways. During the WD season, emissions were consistently dominated by $CH_4$, particularly through ebullition, which remained the primary pathway throughout the entire study period. In contrast, emissions during the WW and CD seasons were largely driven by $CO_2$, with diffusion at the air-water interface serving as the dominant pathway. This seasonal partitioning

underscore the strong influence of thermal stratification and mixing regimes on GHG fluxes in the reservoir.

Degassing of both $CH_4$ and $CO_2$ accounted for only 4% of total $CO_2$eq GHG emissions, though its contribution was more pronounced during the warm seasons due to higher discharges and water turbulence. The relatively minor role of degassing, compared to diffusion and ebullition, suggests that future design of reservoirs should include feature comparable to the one in NT2 reservoir which allows to minimize the downstream emissions (Deshmukh et al., 2016).

Long-term monitoring of the NT2 reservoir revealed a consistent decline in greenhouse gas emissions, due to the slow exhaustion of the pool of labile OM. We hypothesise that the reservoir maturation in the subtropics is faster than in the tropical region due to annual overturn that enhance flooded OM degradation. This trend matches with patterns observed in other tropical and subtropical reservoirs, supporting the notion that emission reductions are a common trajectory as reservoirs mature.

When comparing gross GHG emissions from NT2 with other tropical reservoirs of similar monitoring duration, Petit Saut in

French Guiana offers a relevant benchmark (10 years of measurement, Abril et al., 2005). The long-term trend of GHG emissions from the NT2 reservoir showed a slower decline compared to the Petit Saut reservoir, particularly for $CH_4$. In Petit Saut, total carbon emissions dropped from 0.37 Mt C year$^{-1}$ during the first three years post-impoundment from 1994 to 0.12 Mt C year$^{-1}$ by 2000, representing a reduction of approximately 68% in a decade. In contrast, NT2 showed a more gradual decrease from 1276 Gg $CO_2$eq in 2010 to 389 Gg $CO_2$eq in 2021, or approximately 70% over 12 years, with $CH_4$ remaining a

significant contributor throughout, especially in later period after 2017. Both NT2 and Petit Saut reservoirs exhibited a clear declining trend in greenhouse gas emissions over more than a decade following impoundment, but the rates and pathways of decrease were different. At NT2, $CO_2$ emissions were dominated by diffusive fluxes and declined by approximately 87% from 2011 to 2021 while, at Petit Saut, diffusive $CO_2$ fluxes were also predominant (61%) and decreased by about 65% over 10 years (Abril et al., 2005). For $CH_4$, NT2 emissions were mainly driven by ebullition which remained relatively stable over

time, whereas diffusive $CH_4$ emissions decreased sharply (97% loss from 2010 to 2021). In contrast, Petit Saut showed a rapid early decline in $CH_4$ ebullition, from around 50 mmol m$^{-2}$ d$^{-1}$ in 1994 to approximately 0.7 mmol m$^{-2}$ d$^{-1}$ by 2003, a reduction of over 98% (Abril et al., 2005). $CH_4$ diffusion at Petit Saut also declined significantly and stabilized at low levels



(approximately 2 mmol m$^{-2}$ d$^{-1}$) after 1996. Overall, the CH$_4$ emission decrease at Petit Saut was steeper and occurred earlier, whereas NT2 maintained high ebullition rates throughout the monitoring period (as mentioned in 4.3). These contrasting trends

likely reflect site-specific characteristics, including reservoir morphology (Petit Saut is a deep reservoir, with a maximum depth of 35 m and an average depth of 20–25 m, featuring a deeper water intake compared to NT2), OM composition (approximately 10 Mt C of recalcitrant carbon derived from the woody biomass of dense natural forest at Petit Saut), and differing thermal regimes, such as permanent stratification and less frequent and complete seasonal mixing at Petit Saut (Abril et al., 2005). These factors affected the dynamics of OM degradation, vertical transport and CH$_4$ oxidation efficiency.

**5 Conclusions**

This 14-year investigation of the NT2 reservoir provided a thorough evaluation of seasonal and interannual variations in CH$_4$ and CO$_2$ emissions from a subtropical hydroelectric system, while also comparing measurement techniques to identify the most effective approaches. The findings stressed the importance of integrating high-frequency EC data with discrete sampling to account for both temporal dynamics and spatial heterogeneity in emissions. Notably, EC consistently reported higher CO$_2$

fluxes than discrete sampling, likely reflecting its enhanced sensitivity to near-shore and shallow zones where emissions are elevated. In contrast, CH$_4$ estimates from EC and discrete methods were more closely comparable, though discrepancies persisted, due to differences in spatial coverage and sampling resolution.

The findings demonstrated the strong influence of reservoir stratification, climatic conditions, and organic matter dynamics on GHG fluxes. CH$_4$ emissions, mainly driven by ebullition, peaked during the WD season due to intensified methanogenesis and

reduced hydrostatic pressure, whereas CO$_2$ emissions were highest during the CD season when full water column mixing occurred. Both gases exhibited an overall declining trend over time, reflecting the gradual depletion of labile OM and the natural aging of the reservoir. Despite this, CH$_4$ bubbling remained remarkably stable throughout the 14-year period, sustained by the substantial pool of soil OM. This highlighted the long-term importance of the OM pool in sediment in driving CH$_4$ emissions, particularly as ebullition became an increasingly dominant pathway. Furthermore, the NT2 reservoir's design

effectively minimized degassing to the downstream channel, with this pathway contributing less than 5% of total emissions over the study period.

Finally, the research emphasized the dynamic nature of GHG emissions in subtropical reservoirs. These findings were critical for refining carbon budgets and evaluating the climate impact of hydroelectric projects, particularly in rapidly developing tropical and subtropical regions.

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
