# Peer review of "Variability of greenhouse gas (CH4 and CO2) emissions in a subtropical hydroelectric reservoir: Nam Theun 2 (Lao PDR)"

_EGUsphere, 2025_

## Author Comment (AC1)

**Response to Reviewer 1: Dr. Alex Zavarsky**

Manuscript title: Temporal variability of greenhouse gas (CH4 and CO2) emissions in a

subtropical hydroelectric reservoir: Nam Theun 2 (Lao PDR)

Manuscript number: EGUSPHERE-2025-3295

Journal: Biogeosciences

**Authors:** Anh-Thái Hoàng, Frédéric Guérin, Chandrashekhar Deshmukh, Axay Vongkhamsao, Saysoulinthone Sopraseuth, Vincent Chanudet, Stéphane Descloux, Toan Vu Duc and Dominique

Serça

**Introduction**

We sincerely thank Dr. Alex Zavarsky for their thorough evaluation and supportive remarks regarding the scope, relevance, and quality of our dataset. We appreciate the recognition of our work's contribution to understanding CH4 and CO2 dynamics in hydroelectric reservoirs and the positive assessment of the long-term dataset and methodological diversity (eddy covariance, discrete water sampling, and ebullition funnel measurements).

For the context, this article represents a continuation and advancement of previous research initiated during the early years following the impoundment of the NT2 reservoir, spanning from 2009 to 2012. Earlier works (Deshmukh et al., 2014, 2014, 2018; Guérin et al., 2016; Serça et al., 2016) laid the foundational understanding of GHG dynamics and water quality conditions (Chanudet et al., 2016) within the framework of an extensive monitoring project (Descloux et al., 2016). The primary objective of this process-based article was to provide a comprehensive assessment of gross GHG emissions (CO2 and CH4) with temporal variation and seasonality of GHGs from the NT2 reservoir water. Moreover, the 14-year dataset will also support two additional research contributions. First, it enabled the accurate quantification of net GHG emissions from the reservoir (considering pre-impoundment, and additional pathways such as downstream emission and drawdown area emissions), including N2O fluxes, an important addition, and the application of updated life-cycle analysis methods (Guérin et al, in prep). Second, it contributed to an original study focused on the nitrogen cycle within the NT2 reservoir, marking a significant expansion beyond previous works which centered on carbon dynamics (CO2 and CH4). Collectively, these forthcoming articles, together with the current assessment, aim to provide a comprehensive understanding of biogeochemical processes governing greenhouse gas emissions from tropical reservoirs.

We fully acknowledge the reviewer's valuable suggestions regarding the structure and focus of the *Discussion* (Section 4) and the need for a more integrative and comparative *Conclusion*.

In response, we have undertaken substantial revisions as follows:

- We reorganized the Chapter 4, *Discussion*, as follows:
  - o 4.1 Diurnal variation of GHG emissions
    - 4.1.1 CH4 emission
    - 4.1.2 CO2 emission
  - 4.2 Seasonal variation of GHG emissions
    - 4.2.1 CH4 emission
    - 4.2.2 CO2 emission
  - 4.3 Annual variation of GHG emissions
    - 4.3.1 CH4 emission
    - 4.3.2 CO2 emission
    - 4.3.3 Comparison with the Petit-Saut (French Guyana) Reservoir: Longterm trends in GHG emissions
  - 4.4 Comparison between upscaled eddy-covariance fluxes and estimates from discrete sampling

By this new organization, we emphasized each point of discussion clearer with sub-section according to each GHG (CH4 and CO2), so that the audience can easily follow. Also, we removed section 4.5 as it overlapped with the interannual variation sections, hence we added the comparison with Petit-Saut to section 4.3 as a subsection.

- We improved our key findings as follows:
  - o Diurnal variations:
    - CH4: clear diurnal variation (p <0.05) as daytime fluxes higher than nighttime fluxes
    - CO2: 2010 and 2011 showed no significant different (p=0.84 and 0.80, respectively) between daytime and nighttime fluxes. Only 2009 showed significant difference between daytime and nighttime fluxes (Night > Day, p <0.05). Therefore, diurnal variation of CO2 was only significant during the WD season as May 2009 was the case study (2010, 2011 campaigns were in March, late CD season and early WD season).

 While our estimates were considered robust, they should be interpreted as conservative due to the uncorrected diurnal variability.

**Comparison of methods:**

- EC measurements provided continuous, high-temporal-resolution data, allowing capture of short-term events (for example, peaks in flux) and diurnal cycles, including nighttime emissions.
- In contrast, discrete sampling captured spatial variability across the reservoir, but was limited to daytime measurements.
- In this study, the discrete sampling dataset was considered more representative for overall emission calculations because it provided denser spatial coverage and higher data availability across the reservoir. Therefore, emission estimates reported in the article were primarily based on the discrete sampling approach, while EC data were used as complementary to support temporal dynamics and highlight short-term variability.

**Emissions:**

- CH4 and CO2 fluxes exhibited distinct seasonal patterns linked to their dominant emission pathways:
  - CH4 emissions peaked during the WD season, primarily driven by ebullition
  - CO2 emissions peak during the CD season, dominated by diffusive fluxes associated with water column overturn, which brings CO2-rich deeper water to the surface.
  - Drivers of GHG fluxes: Seasonal and short-term variations in CH4 and CO2 emissions are controlled by a combination of reservoir stratification, hydrological dynamics (e.g., water level fluctuations), and meteorological factors (e.g., temperature, wind, and precipitation). These factors influence the strength of diffusive fluxes, bubble formation, and water—air gas exchange.
  - Long-term trends: Over the 14-year monitoring period, diffusive emissions of both CH4 and CO2 have declined. In contrast, CH4

emissions from ebullition have remained relatively stable, sustained by the availability of OM pools in flooded sediments and vegetation.

 In our next article, we calculated that the carbon inputs from the watershed, 34 GgC year-1 on average, could have contributed to only 15% of total CO2 and CH4 emissions (Guérin, Deshmukh, Hoàng et al., to be submitted)

Those conclusions are more concise and emphasized well the key findings of the article.

Below, we would like to answer to your point-to-point questions:

Line 55 and after: There are three mechanisms: ebullition, diffusive fluxes and degassing. One could briefly explain what these three are and how they are measured. Ebullitionà Bubble traps. Diffusive fluxes EddyCov, K\*DeltaC, ebullition upstream-downstream.

*Answer*: I added the definition of each pathways only. The methods used to collect them were mentioned in the Material and methods. (Line 55)

**Line 88: The Dam was impounded and the commissioned.**

*Answer:* Yes, the impoundment of the reservoir started in 04/2008 and fully commissioned in 04/2010.

**Was happened when it was commissioned? Water through the turbines?**

Answer: Yes, the turbines started in 03/2010

**Was the water before discharged via the spill-over?**

*Answer:* From 04/2008 to 03/2010, the water was discharged through the dams and spill-way (in the event of flood)

**Line 250: What are gross emissions. Is it already source and sink subtracted? Is it influx of OM minus GHG coming out?**

Answer: Gross emissions are those that are directly measurable from existing reservoirs (Rasanen et al., 2018), which means that it accounts only for the amount of emissions after the reservoir is stably impounded (2009 onwards), and from the water body only. The net emission of NT2, considering also the emissions from the reservoir area before inundation, which can act as a GHG source (e.g. natural waters) or sink (e.g. forests) (Rasanen et al., 2018), which will be reported in a separate article with more pathways and the downstream structures. I added this definition to the

Line 505-510: The ebullition effect of atmospheric pressure change. Did you see this also in the EC data?

Answer: Yes, the EC measurement captures both diffusive fluxes and ebullitive fluxes from the water surface (Deshmukh et al., 2014)

Line556: GE measurements (TBL and bubbling) is this the calculation method for Gross Emissions? This should be explained before.

Answer: I added a term DE (diffusion + ebullition) instead of GE for the comparison with EC to clarify. Also, I clarified the terminology in Section 2.9 and Section 2.12

Line 591: In the methods section there should be a clear definition of EC and TBL(GE) method. Then just use one abbreviation TBL or GE. I think that the way of calculating GE is through TBL. That should not be mixed up.

Answer: I changed the term to DE when it comes to the comparison with EC.

Line 608: kt values are often highly discussed and vary regarding which parametrization you use. This could be mentioned earlier when you compare the fluxes.

Answer: the kt value is the important components of the calculation of diffusive fluxes. Since the first fluxes comparison presented in the diurnal variation only used the EC direct measurement data, hence the kt did not play an important role in this comparison. For CH4 fluxes, it consisted of both bubbling and diffusion, and from the results showed in Section 3.3, in 2022, around 95% of the CH4 rooted from bubbling. Hence the kt value is most significant when it comes to diffusive CO2 fluxes which are the main source of CO2 emissions and about 40% of the total GE.

Line 617: "Temporal dynamics of CH4 emissions from the reservoir water surface." Why is the abstract called "from the water surface" you mention diffusive fluxes, ebullition and degassing and water discharged at the pill-over. What is so significant to the water surface now? You describe EC before and now the other pathways. I would choose a different subtitle.

Answer: This paper strictly quantified the amount of CO2 and CH4 from the water body / water column of the main reservoir NT2, disregarding the emissions from the drawdown area (soil) and the downstream emissions (after the turbines). I will give a clearer answer below. About the title, I have reorganized the whole Section 4 discussion to be clearer and easier to follow for the audiences.

Line 631: Do you mean the water is discharge from the reservoir or coming from the surroundings into the reservoir?

*Answer:* I referred to the amount of DOC and IC coming from the watershed (tributaries) into the reservoir during the warm season.

Line 780: Degassing + Feature. It should be written that the relative minor role of degassing due to the features at the damn, suggests that future projects....

Answer: I added the information related to the design of NT2 reservoir: "such as intake configuration or artificial mixing system, which introduces destratification and oxygenation, upstream of the turbines"

Line 790: This is a very interesting paragraph putting your measurements in relation to others.

Answer: Thank you very much for this comment.

Line 824: Can you just briefly remind us what the design features are: intake depth, ventilation, ... just one or two catchwords.

Answer: I added the information: "the introduction of artificial mixing of the water column before the turbines"

General question: Did you take a deeper look at water-level influence. You mention the hydrostatic pressure changes influencing ebullition but only cite Deshmukh 2014. Have you seen any influence from falling dry and resubmerged banks?

Answer: This article focused mainly on the GHG emissions (CO2 and CH4) from the water body and the corresponding pathways such as diffusion, ebullition, and degassing directly from the main

reservoir (turbines, dam). The paper focused mostly on the processes underlaying the fluxes, which will contribute and compliment to another upcoming paper by our team reporting the broader view of net emissions which includes drawdown emissions (from the bank with the water variation effect as reported by Serca et al., 2016), downstream emissions (from the downstream structure of the NT2 reservoir, as recently reported in Deshmukh et al., 2016), the pre- and post-impoundment net balance and life cycle analysis (LCA). The results and discussions of this article will be fundamental to the net emission paper, as well as another article later for nitrogen circle and N2O emissions. From this impressive 14-year database of measurements, our team will provide the comprehensive reports on long-term emissions as well as biogeochemistry processes underlying those emissions.

Technical remarks: I completed all the minor changes and included them to the final versions.

Abstract Line 31: I would spell it out in the abstract "warm dry". The same goes for cold dry. They are not too long. Its perfectly fine in the main body – Done.

Line 43: At the first mentioning I would write greenhouse gas (GHG) – Done.

Line 470/ Figure 5: You use ebullition and bubbling. I would recommend only use ebullition. This also would be appropriate for figures. – Done, I changed to "bubbling" to match with the figure legend. – Done

**References**

Chanudet, V., Fabre, V., and van der Kaaij, T.: Application of a three-dimensional hydrodynamic model to the Nam Theun 2 Reservoir (Lao PDR), Journal of Great Lakes Research, 38, 260–269, https://doi.org/10.1016/j.jglr.2012.01.008, 2012.

Chanudet, V., Smits, J., Van Beek, J., Boderie, P., Guérin, F., Serça, D., Deshmukh, C., and Descloux, S.: Hydrodynamic and water quality 3D modelling of the Nam Theun 2 Reservoir (Lao PDR): predictions and results of scenarios related to reservoir management, hydrometeorology and nutrient input, Hydroécol. Appl., 19, 87–118, https://doi.org/10.1051/hydro/2014009, 2016.

Descloux, S., Chanudet, V., Taquet, B., Rode, W., Guédant, P., Serça, D., Deshmukh, C., and Guerin, F.: Efficiency of the Nam Theun 2 hydraulic structures on water aeration and methane degassing, Hydroécol. Appl., 19, 63–86, https://doi.org/10.1051/hydro/2015002, 2016a.

Descloux, S., Guedant, P., Phommachanh, D., and Luthi, R.: Main features of the Nam Theun 2 hydroelectric project (Lao PDR) and the associated environmental monitoring programmes, Hydroécol. Appl., 19, 5–25, https://doi.org/10.1051/hydro/2014005, 2016b.

Deshmukh, C., Serça, D., Delon, C., Tardif, R., Demarty, M., Jarnot, C., Meyerfeld, Y., Chanudet, V., Guédant, P., Rode, W., Descloux, S., and Guérin, F.: Physical controls on CH4 emissions from a newly flooded subtropical freshwater hydroelectric reservoir: Nam Theun 2, Biogeosciences, 11, 4251–4269, https://doi.org/10.5194/bg-11-4251-2014, 2014.

Deshmukh, C., Guérin, F., Labat, D., Pighini, S., Vongkhamsao, A., Guédant, P., Rode, W., Godon, A., Chanudet, V., Descloux, S., and Serça, D.: Low methane (CH4) emissions downstream of a monomictic subtropical hydroelectric reservoir (Nam Theun 2, Lao PDR), Biogeosciences, 13, 1919–1932, https://doi.org/10.5194/bg-13-1919-2016, 2016.

Deshmukh, C., Guérin, F., Vongkhamsao, A., Pighini, S., Oudone, P., Sopraseuth, S., Godon, A., Rode, W., Guédant, P., Oliva, P., Audry, S., Zouiten, C., Galy-Lacaux, C., Robain, H., Ribolzi, O., Kansal, A., Chanudet, V., Descloux, S., and Serça, D.: Carbon dioxide emissions from the flat bottom and shallow Nam Theun 2 Reservoir: drawdown area as a neglected pathway to the atmosphere, Biogeosciences, 15, 1775–1794, https://doi.org/10.5194/bg-15-1775-2018, 2018.

Guérin, F., Deshmukh, C., Labat, D., Pighini, S., Vongkhamsao, A., Guédant, P., Rode, W., Godon, A., Chanudet, V., Descloux, S., and Serça, D.: Effect of sporadic destratification, seasonal overturn, and artificial mixing on CH4 emissions from a subtropical hydroelectric reservoir, Biogeosciences, 13, 3647–3663, https://doi.org/10.5194/bg-13-3647-2016, 2016.

Serça, D., Deshmukh, C., Pighini, S., Oudone, P., Vongkhamsao, A., Guédant, P., Rode, W., Godon, A., Chanudet, V., Descloux, S., and Guérin, F.: Nam Theun 2 Reservoir four years after commissioning: significance of drawdown methane emissions and other pathways, Hydroécol. Appl., 19, 119–146, https://doi.org/10.1051/hydro/2016001, 2016.

---

## Author Comment (AC2)

**Response to Reviewer 2**

Manuscript title: Temporal variability of greenhouse gas (CH4 and CO2) emissions in a

subtropical hydroelectric reservoir: Nam Theun 2 (Lao PDR)

Manuscript number: EGUSPHERE-2025-3295

**Journal:** Biogeosciences

**Authors:** Anh-Thái Hoàng, Frédéric Guérin, Chandrashekhar Deshmukh, Axay Vongkhamsao, Saysoulinthone Sopraseuth, Vincent Chanudet, Stéphane Descloux, Toan Vu Duc and Dominique

Serça

**Introduction**

We sincerely thank Reviewer #2 for their careful evaluation of our manuscript and for recognizing the value of the long-term and seasonally resolved dataset from this hydroelectric reservoir. We appreciate the reviewer's constructive feedback regarding the clarity and focus of the Results and Discussion sections.

For the context, this article represents a continuation and advancement of previous research initiated during the early years following the impoundment of the NT2 reservoir, spanning from 2009 to 2012. Earlier works (Deshmukh et al., 2014, 2014, 2018; Guérin et al., 2016; Serça et al., 2016) laid the foundational understanding of GHG dynamics and water quality conditions (Chanudet et al., 2016) within the framework of an extensive monitoring project (Descloux et al., 2016). The primary objective of this process-based article was to provide a comprehensive assessment of gross GHG emissions (CO2 and CH4) with temporal variation and seasonality of GHGs from the NT2 reservoir water. Moreover, the 14-year dataset will also support two additional research contributions. First, it enabled the accurate quantification of net GHG emissions from the reservoir (considering pre-impoundment, and additional pathways such as downstream emission and drawdown area emissions), including N2O fluxes, an important addition, and the application of updated life-cycle analysis methods (Guérin et al, in prep). Second, it contributed to an original study focused on the nitrogen cycle within the NT2 reservoir, marking a significant expansion beyond previous works which centered on carbon dynamics (CO2 and CH4). Collectively, these forthcoming articles, together with the current assessment, aim to provide a comprehensive understanding of biogeochemical processes governing greenhouse gas emissions from tropical reservoirs.

We fully acknowledge the reviewer's valuable suggestions regarding the structure and focus of the *Discussion* (Section 4) and the need for a more integrative and comparative *Conclusion*.

In response, we have undertaken substantial revisions as follows:

- We reorganized the Chapter 4, *Discussion*, as follows:
  - o 4.1 Diurnal variation of GHG emissions
    - 4.1.1 CH4 emission
    - 4.1.2 CO2 emission
  - 4.2 Seasonal variation of GHG emissions
    - 4.2.1 CH4 emission
    - 4.2.2 CO2 emission
  - 4.3 Annual variation of GHG emissions
    - 4.3.1 CH4 emission
    - 4.3.2 CO2 emission
    - 4.3.3 Comparison with the Petit-Saut (French Guyana) Reservoir: Longterm trends in GHG emissions
  - 4.4 Comparison between upscaled eddy-covariance fluxes and estimates from discrete sampling

By this new organization, we emphasized each point of discussion clearer with sub-section according to each GHG (CH4 and CO2), so that the audience can easily follow. Also, we removed section 4.5 as it overlapped with the interannual variation sections, hence we added the comparison with Petit-Saut to section 4.3 as a subsection.

- We improved our key findings as follows:
  - o Diurnal variations:
    - CH4: clear diurnal variation (p <0.05) as daytime fluxes higher than nighttime fluxes
    - CO2: 2010 and 2011 showed no significant different (p=0.84 and 0.80, respectively) between daytime and nighttime fluxes. Only 2009 showed significant difference between daytime and nighttime fluxes (Night > Day, p <0.05). Therefore, diurnal variation of CO2 was only significant during the WD season as May 2009 was the case study (2010, 2011 campaigns were in March, late CD season and early WD season).
    - While our estimates were considered robust, they should be interpreted as conservative due to the uncorrected diurnal variability.

**o Comparison of methods:**

- EC measurements provided continuous, high-temporal-resolution data, allowing capture of short-term events (for example, peaks in flux) and diurnal cycles, including nighttime emissions.
- In contrast, discrete sampling captured spatial variability across the reservoir, but was limited to daytime measurements.
- In this study, the discrete sampling dataset was considered more representative for overall emission calculations because it provided denser spatial coverage and higher data availability across the reservoir. Therefore, emission estimates reported in the article were primarily based on the discrete sampling approach, while EC data were used as complementary to support temporal dynamics and highlight short-term variability.

**o Emissions:**

- CH4 and CO2 fluxes exhibited distinct seasonal patterns linked to their dominant emission pathways:
  - CH4 emissions peaked during the WD season, primarily driven by ebullition
  - CO2 emissions peak during the CD season, dominated by diffusive fluxes associated with water column overturn, which brings CO2rich deeper water to the surface.
  - Drivers of GHG fluxes: Seasonal and short-term variations in CH4 and CO2 emissions are controlled by a combination of reservoir stratification, hydrological dynamics (e.g., water level fluctuations), and meteorological factors (e.g., temperature, wind, and precipitation). These factors influence the strength of diffusive fluxes, bubble formation, and water—air gas exchange.
  - Long-term trends: Over the 14-year monitoring period, diffusive emissions of both CH4 and CO2 have declined. In contrast, CH4 emissions from ebullition have remained relatively stable, sustained by the availability of OM pools in flooded sediments and vegetation.

 In our next article, we calculated that the carbon inputs from the watershed, 34 GgC year-1 on average, could have contributed to only 15% of total CO2 and CH4 emissions (Guérin, Deshmukh, Hoàng et al., to be submitted)

Those conclusions are more concise and emphasized well the key findings of the article.

Below, we would like to answer to your point-to-point questions:

Presentation of results and discussion – Much of the Results and Discussion are very long and detailed, and distract from the most important findings of the paper. For example, Section 3.2 in the Results takes up a lot of space for being only ancillary to the main findings of this paper. I suggest significantly reducing this section and perhaps putting it at the beginning of the Results (details could go in supplementary materials). The Results could then begin with a very brief overview of these vertical chemical dynamics, and follow with emphasis on the emissions comparisons between methods and over time, which are the real highlights of this paper. Similarly, the Discussion is very long winded, with many sections reading like a laundry list of similar studies without clearly linking to key findings of this study. I suggest focusing on linking specific results with mechanisms and relevant papers, and highlighting the specific, novel contributions from this study. Some closing paragraphs for long sections (i.e., paragraphs beginning on line 699 and 766) could succinctly summarize much of the section without needing multiple pages of text.

Answer: thank you for this suggestion. The *Result* sections, especially the 3.2 about vertical profiles are essential for our theme of research in emissions of the NT2 reservoir. This article served as a process-based article to publish results that will be linked to a net-emission and carbon transfer article (Guérin et al., to be submitted). Hence, we would like to present the trends and values of each carbon species in details, together with mixing indicators (water temperature and dissolved oxygen)

The *Discussion* was reorganized and reduced as mentioned above with more precise conclusions for each section.

• Integrating statistical tests – Much of the Results section uses visual or tabular comparisons to make statements about differences in emissions rates, when statistical tests should explicitly be used. For example, comparisons of GE versus EC emissions are only presented using mean and uncertainty, when t-tests could easily compare these rates across all years.

The authors write in line 349 emissions rate from EC were significantly higher than GE in some campaigns, including in March 2011 as  $1.31 \pm 0.10$  Gg CH4 month-1 (EC) versus 1.25  $\pm$  0.23 Gg CH4 month-1 (GE). However, considering the uncertainty values, these are not really different than each other. Please incorporate appropriate statistical tests for these and other comparisons (i.e., statistical trend tests for results in lines 486-492; mechanistic statement in line 620) rather than only qualitatively or visually describing results.

Answer: We performed additional statistical tests on diurnal variation and EC comparison. For the prior results, only CH4 showed clear diurnal variation (p<0.05, Night < Day in terms of flux magnitudes), while CO2 showed diurnal variation only in the WD seasons (2009, p<0.05, Night > Day in term of flux magnitudes), while March 2010 and March 2011 showed no differences between Day and Night fluxes (p=0.84 and 0.80, respectively).

For the comparison, all of  $CH_4$  comparison were significantly different (p <0.05), almost the same statistical test results showed for  $CO_2$ , only 2011 campaigns showed similarity between the two (p = 0.47).

We have done statistical tests on all emission results (trend; seasonal, interannual variations). We will incorporate those statistical results to support our claims, such as line 486-492 (p<0.05).

Upscaling from few stations to entire reservoir – The 3-7 sites used for upscaling emissions rates to the whole-reservoir scale seem very few to precisely capture the full range in spatial variability in emissions rates. How does this number of sites compare to recommendations in Beaulieu et al. 2016 (https://doi.org/10.1002/lno.10284, see Fig. 5) or to the area covered by sampling each site in previous work (i.e., **Jager** et al. 2022, https://doi.org/10.1016/j.rser.2022.112408, see Fig. 6)? The few numbers of sites is a major caveat for this scale of whole-reservoir upscaling and needs more attention and/or Discussion.

Answer: The Nam Theun 2 reservoir hydrological assessment was published by Chanudet et al., 2012 (doi:10.1016/j.jglr.2012.01.008) mentioned due to their special hydrological conditions that RES9 is accounted for 3 km2 before the turbine, while RES3 is accounted for 5.5% of the total area regardless of the seasonal fluctuation of water level.

The other 7 stations (RES1, RES2, RES4-8) were statistically tested for diffusive fluxes (CH4 and CO2) to see if they were spatially different. The results showed that they were statistically similar for all 14 years of measurement. Hence, they represented the rest of the reservoir surface (100% - 5.5% - 3 km2). These results were similar to those which have been reported by Deshmukh et al., (2014, 2018)

and Guérin et al. (2016) in NT2 reservoir during the period 2009-2013. We included this information in the Section 2.5 Total diffusive fluxes calculation as special spatial variability test.

• Linking proposed mechanisms to observed patterns – There are several instances in the Discussion where mechanisms are briefly proposed, but have limited statistical backing or additional explanations that are omitted. For example, in line 711, the measured CO2 concentrations represent the net remaining between CO2 consumption and uptake/emissions, not specifically the total CO2 Hence, the production of CO2 in the surface waters is likely more readily respired or emitted resulting in lower CO2 concentrations, which is not specifically a reflection of the gross CO2 production rates at depth. Another example in line 781 omits the accumulation of gases in deep water during the stratified period as a direct connection with higher degassing emissions during the warm stratified period. In general, more thorough mechanistic linkages would be warranted while removing much of the text that is not as relevant to the key findings of this study.

Answer: Thank you very much for this suggestion.

Line 711: We changed from consumed by bacteria to consumed by phytoplankton as phytoplankton biomass increased during the WD season.

Line 781: In Nam Theun 2, degassing was reduced significantly due to the design of this reservoir. Before the turbine (RES9), we introduced an artificial mixing that outgassed both gases before downstream emissions. Before the dam (RES1), even though deep GHG-rich water going through the dam, but the discharge was only 2 m3 s-1 constantly. The high fluxes came only in the event of flood (spill-way release) but those event were rare.

We will carefully review the texts and mechanism behind fluxes.

**Minor comments:**

Line 56 – Disregarding degassing emissions can reduce emissions by a large portion (see, for example, Harrison et al. 2021, https://doi.org/10.1029/2020GB006888). Since you calculate and present degassing emissions, I suggest incorporating this into this rough breakdown of emissions by pathway.

Answer: In addition to the previous remark on the degassing emissions, in Nam Theun 2 degassing only consisted of 4% of total emissions, and was stable through 14 years of study (p > 0.05). The degassing process of NT2 was previously reported as minor pathway in Deshmukh et al., 2016

(doi:10.5194/bg-13-1919-2016) for CH4 and will be reported again in Guérin et al., (to be submitted) for CH4, and CO2.

• Line 58 – Consider here (and elsewhere as relevant) to discuss potential CO2 uptake (i.e., CO2 influx into the reservoir rather than emissions) that can be common in productive systems, as well as the variability in uptake versus emissions at diurnal to seasonal scales. This is particularly important given the negative CO2 rates presented in Table 1.

Answer: We had calculated the discharge of carbon and nutrient into the reservoir, and will be presented in next articles by Guérin et al., (to be submitted) on Net emissions and nitrogen cycling. The result showed that the contribution of the influx discharge to the reservoir contributed to only 15% of the total emissions.

• Line 75 – An eddy covariance flux tower is unlikely to be particularly useful for capturing spatial variability, as it integrates signal across a specific area without distinguishing variability in those signals within that area. Many would need to be set up around a reservoir to really get at spatial variability effectively. Given the scope of this paper with one EC tower set up, I suggest reducing the emphasis on capturing spatial variability via this method.

Answer: Yes, we emphasized the EC continuous measurements are able to capture diurnal variation (nighttime fluxes) and short-event (peaks) in the discussion. Hence, we will rephrase this line to temporal variation only.

• Line 146 – While ebullition does generally occur in shallower areas, it is possible to have ebullition occurring deeper than 16 m. I suggest rephrasing this and incorporating a couple of references that show ebullition more dominant – but not exclusive to – shallower areas in reservoirs (i.e., DelSontro et al. 2011, https://dx.doi.org/10.1021/es2005545; Beaulieu et al. 2016, https://doi.org/10.1002/lno.10284).

*Answer:* In the NT2, we did not observe any CH4 bubbling further than 16m of depth. This is case sensitive, therefore, we will rephrase it to specifically mention NT2.

• Line 243 – Can you add more details on the ANN approach for the bubbling emissions estimates? This is a highly variable pathway over space and time, so additional details here (or in supplementary material) would be useful to describe model fitting and performance.

Answer: In this study, an artificial neural network was used to find the best non-linear regression between ebullition fluxes and relevant environmental variables. We applied ANN using the 'nnet' package (https://cran.r-project.org/web/packages/nnet/index.html) to model bubbling fluxes. Water

depth, change in water level, , atmospheric pressure, change in atmospheric pressure, total static pressure, change in total static pressure, and reservoir bottom temperature data were used as explanatory variables - see the Deshmukh et al., 2014 for the choice of the inputs. The database of raw data was composed of 6,158 individual ebullition fluxes resulted from about 13 years of measurements (2009 – 2022). Fluxes from a given station measured the same day and at the same depth were averaged (different fluxes with the same depth value and the same meteorological data would introduce noise rather than relevant information into the network), leading to a final input data for ANN composed of 6158 lines and 8 columns (1 output and 7 inputs). The data set is separated into two pools, the training one (80% out of total input data) and the validation one (20% out of total input data). The repeated cross-validation with 10 folds and 5 repetitions were applied to evaluate the performance of the ANN model. Additionally, the ANN model was iterated for 20 times. Averages of the 20 modelled values were used to estimate daily bubbling fluxes and the standard deviation was used to quantify the uncertainty in the estimates, the overall model performance reached up to 66% on the daily time scale.

• Line 277 – Can you elaborate on this "spike removal process" for EC data processing? Given the often pulsed emissions nature of ebullition, it might be expected to find occasional to frequent "spikes" in CH4 emissions as a result. Does this data processing remove those and hence artificially reduce the accuracy of ebullitive emissions variability?

Answer: The "spike removal process" applied in our study followed a standard de-spiking procedure, in which both wind vector components and scalar exceeding three times the standard deviation from the local mean were flagged and removed.

This approach is widely used in EC studies to eliminate physically implausible or instrument-related outliers, while retaining the majority of real variability.

We acknowledge that CH4 emissions from ebullition are often pulsed, and care must be taken to avoid artificially removing genuine high-flux events. In our implementation: The algorithm removes only extreme outliers that exceed the 3 $\sigma$  threshold, which primarily correspond to measurement artifacts or spikes unrelated to physical emission events. Short-term pulsed ebullition events that fall within the expected natural variability are retained, ensuring that the EC data continue to reflect the true temporal dynamics of CH4 fluxes.

• Line 320 – What specific results is this calculation of SD and SE used for? If used for whole-reservoir upscaled results (i.e., from the 3-7 measurement sites), consider utilizing

propagated error based on individual measurement uncertainty and portion of the reservoir each represents.

Answer: The 7 measurement sites represented an equal proportion of the reservoir after exclude RES9 and RES3.

The SE was used to take into consideration the number of fluxes that made up the mean.

• Line 338 – How comparable are the EC results from 2009-2011 versus 2019 and 2022, given the EC flux tower was (1) located in a very different part of the reservoir between these time periods, and (2) data span different months/seasons?

Answer: We tested those results statistically, and they showed significant difference. This result are consistence with the finding that emissions declined with time for both gases CH4 and CO2 and they both showed distinct seasonal variations.

• Line 341 – Suggest adding a figure (in the main text or supplementary) showing the differences between daytime and nighttime emissions patterns, as it is a major emphasis in the manuscript.

*Answer:* In my previous PhD defense, I acknowledged this comment and showed the two figures as shown below. We will discuss to decide to include those into the main text or to the supplementary. I believe they are good additions.

• Figure 3 – Suggest adding error bars to each estimate to visualize uncertainty (i.e., as in Figure 5), which would pair with explicit statistical tests as suggested above.

Answer: Yes, I added the error bars to the two graphs as shown in the example below.

• Figure 4 – This figure is quite difficult to fully grasp. Consider reducing the number of panels/variables to include only those that are most different or relevant to the emissions patterns (moving other to supplemental material), and/or adding color/different line types to more clearly distinguish the different years. At its current size, the different symbols used are impossible to distinguish.

Answer: We could add color to our graphs, but it depends on the APC for colored article. I learned that the APC rate now is flat for full article, whether colored or black-white resolutions. However, we need to check carefully on this matter. I attached some colored graphs for your references.

Evolution of greenhouse gas emissions from 2009 to 2022 by gases

**References**

Chanudet, V., Fabre, V., and van der Kaaij, T.: Application of a three-dimensional hydrodynamic model to the Nam Theun 2 Reservoir (Lao PDR), Journal of Great Lakes Research, 38, 260–269, https://doi.org/10.1016/j.jglr.2012.01.008, 2012.

Chanudet, V., Smits, J., Van Beek, J., Boderie, P., Guérin, F., Serça, D., Deshmukh, C., and Descloux, S.: Hydrodynamic and water quality 3D modelling of the Nam Theun 2 Reservoir (Lao PDR): predictions and results of scenarios related to reservoir management, hydrometeorology and nutrient input, Hydroécol. Appl., 19, 87–118, https://doi.org/10.1051/hydro/2014009, 2016.

Descloux, S., Chanudet, V., Taquet, B., Rode, W., Guédant, P., Serça, D., Deshmukh, C., and Guerin, F.: Efficiency of the Nam Theun 2 hydraulic structures on water aeration and methane degassing, Hydroécol. Appl., 19, 63–86, https://doi.org/10.1051/hydro/2015002, 2016a.

Descloux, S., Guedant, P., Phommachanh, D., and Luthi, R.: Main features of the Nam Theun 2 hydroelectric project (Lao PDR) and the associated environmental monitoring programmes, Hydroécol. Appl., 19, 5–25, https://doi.org/10.1051/hydro/2014005, 2016b.

Deshmukh, C., Serça, D., Delon, C., Tardif, R., Demarty, M., Jarnot, C., Meyerfeld, Y., Chanudet, V., Guédant, P., Rode, W., Descloux, S., and Guérin, F.: Physical controls on CH4 emissions from a newly flooded subtropical freshwater hydroelectric reservoir: Nam Theun 2, Biogeosciences, 11, 4251–4269, https://doi.org/10.5194/bg-11-4251-2014, 2014.

Deshmukh, C., Guérin, F., Labat, D., Pighini, S., Vongkhamsao, A., Guédant, P., Rode, W., Godon, A., Chanudet, V., Descloux, S., and Serça, D.: Low methane (CH4) emissions downstream of a monomictic subtropical hydroelectric reservoir (Nam Theun 2, Lao PDR), Biogeosciences, 13, 1919–1932, https://doi.org/10.5194/bg-13-1919-2016, 2016.

Deshmukh, C., Guérin, F., Vongkhamsao, A., Pighini, S., Oudone, P., Sopraseuth, S., Godon, A., Rode, W., Guédant, P., Oliva, P., Audry, S., Zouiten, C., Galy-Lacaux, C., Robain, H., Ribolzi, O., Kansal, A., Chanudet, V., Descloux, S., and Serça, D.: Carbon dioxide emissions from the flat bottom and shallow Nam Theun 2 Reservoir: drawdown area as a neglected pathway to the atmosphere, Biogeosciences, 15, 1775–1794, https://doi.org/10.5194/bg-15-1775-2018, 2018.

Guérin, F., Deshmukh, C., Labat, D., Pighini, S., Vongkhamsao, A., Guédant, P., Rode, W., Godon, A., Chanudet, V., Descloux, S., and Serça, D.: Effect of sporadic destratification, seasonal overturn, and artificial mixing on CH4 emissions from a subtropical hydroelectric reservoir, Biogeosciences, 13, 3647–3663, https://doi.org/10.5194/bg-13-3647-2016, 2016.

Serça, D., Deshmukh, C., Pighini, S., Oudone, P., Vongkhamsao, A., Guédant, P., Rode, W., Godon, A., Chanudet, V., Descloux, S., and Guérin, F.: Nam Theun 2 Reservoir four years after commissioning: significance of drawdown methane emissions and other pathways, Hydroécol. Appl., 19, 119–146, https://doi.org/10.1051/hydro/2016001, 2016.

---

## Author Response (AR1)

**Response to Reviewer 1: Dr. Alex Zavarsky**

**Manuscript title:** Temporal variability of greenhouse gas (CH₄ and CO₂) emissions in a subtropical hydroelectric reservoir: Nam Theun 2 (Lao PDR)
**Manuscript number:** EGUSPHERE-2025-3295
**Journal:** *Biogeosciences*

**Authors:** Anh-Thái Hoàng, Frédéric Guérin, Chandrashekhar Deshmukh, Axay Vongkhamsao, Saysoulinthone Sopraseuth, Vincent Chanudet, Stéphane Descloux, Nurholis Nurholis, Ari Putra Susant, Toan Vu Duc and Dominique Serça
* * *
**Introduction**

We sincerely thank Dr. Alex Zavarsky for their thorough evaluation and supportive remarks regarding the scope, relevance, and quality of our dataset. We appreciate the recognition of our work's contribution to understanding CH₄ and CO₂ dynamics in hydroelectric reservoirs and the positive assessment of the long-term dataset and methodological diversity (eddy covariance, discrete water sampling, and ebullition funnel measurements).

For the context, this article represents a continuation and advancement of previous research initiated during the early years following the impoundment of the NT2 reservoir, spanning from 2009 to 2012. Earlier works (Deshmukh et al., 2014, 2014, 2018; Guérin et al., 2016; Serça et al., 2016) laid the foundational understanding of GHG dynamics and water quality conditions (Chanudet et al., 2016) within the framework of an extensive monitoring project (Descloux et al., 2016). The primary objective of this process-based article was to provide a comprehensive assessment of gross GHG emissions (CO₂ and CH₄) with temporal variation and seasonality of GHGs from the NT2 reservoir water. Moreover, the 14-year dataset will also support two additional research contributions. First, it enabled the accurate quantification of net GHG emissions from the reservoir (considering pre-impoundment, and additional pathways such as downstream emission and drawdown area emissions), including N₂O fluxes, an important addition, and the application of updated life-cycle analysis methods (Guérin et al, in prep). Second, it contributed to an original study focused on the nitrogen cycle within the NT2 reservoir, marking a significant expansion beyond previous works which centered on carbon dynamics (CO₂ and CH₄). Collectively, these forthcoming articles, together with the current assessment, aim to provide a comprehensive understanding of biogeochemical processes governing greenhouse gas emissions from tropical reservoirs.

We fully acknowledge the reviewer's valuable suggestions regarding the structure and focus of the *Discussion* (Section 4) and the need for a more integrative and comparative *Conclusion*. In response, we have undertaken substantial revisions as follows:

- We reorganized the Chapter 4, *Discussion*, as follows:
  - 4.1 Methodological comparison: evaluating the differences between EC and manual discrete sampling
  - 4.2 Diurnal variation of fluxes from EC measurement
    - 4.2.1 $CH_4$ fluxes
    - 4.2.2 $CO_2$ fluxes
  - 4.3 Seasonal variation of GHG emissions
    - 4.3.1 $CH_4$ emission
    - 4.3.2 $CO_2$ emission
  - 4.4 Annual variation of GHG emissions
    - 4.4.1 $CH_4$ emission
    - 4.4.2 $CO_2$ emission
    - 4.4.3 Comparison with the Petit-Saut (French Guyana) Reservoir: Long-term trends in GHG emissions

By this new organization, we emphasized each point of discussion clearer with sub-section according to each GHG ($CH_4$ and $CO_2$), so that the audience can easily follow. Also, we removed section 4.5 as it overlapped with the interannual variation sections, hence we added the comparison with Petit-Saut to section 4.4 as a subsection (4.4.3).

- We improved our key findings as follows:
  - Diurnal variations:
    - $CH_4$: clear diurnal variation ($p < 0.05$) as daytime fluxes higher than nighttime fluxes
    - $CO_2$: In 2010 and 2011, there was no significant difference between daytime and nighttime $CO_2$ fluxes ($p = 0.84$ and $0.80$, respectively). In contrast, in 2009 nighttime fluxes were significantly higher than daytime fluxes (Night > Day, $p < 0.05$). Thus, a clear diurnal variation in $CO_2$ was observed only during the WD season in 2009, when measurements were conducted in May.

The 2010 and 2011 campaigns took place in March, during the late CD and early WD seasons, when no significant diurnal pattern was detected.

- While our estimates were considered robust, they should be interpreted as conservative due to the uncorrected diurnal variability.

o Comparison of methods:

- EC measurements provided continuous, high-temporal-resolution data, allowing capture of short-term events (for example, peaks in flux), process studies such as the impacts of changing water level or radiation on fluxes; and diurnal cycles, including nighttime emissions.
- In contrast, discrete sampling captured spatial variability across the reservoir, but was limited to daytime measurements.
- In this study, the discrete sampling dataset was considered more representative for overall emission calculations because it provided denser spatial coverage and higher data availability across the reservoir. Therefore, emission estimates reported in the article were primarily based on the discrete sampling approach, while EC data were used as complementary to support temporal dynamics and highlight short-term variability.

o Emissions:

- $CH_4$ and $CO_2$ fluxes exhibited distinct seasonal patterns linked to their dominant emission pathways:
  - $CH_4$ emissions peaked during the WD season, primarily driven by ebullition
  - $CO_2$ emissions peak during the CD season, dominated by diffusive fluxes associated with water column overturn, which brings $CO_2$-rich deeper water to the surface.
  - Drivers of GHG fluxes: Seasonal and short-term variations in $CH_4$ and $CO_2$ emissions are controlled by a combination of reservoir stratification, hydrological dynamics (e.g., water level fluctuations), and meteorological factors (e.g., temperature, wind, and precipitation). These factors influence the strength of diffusive fluxes, bubble formation, and water–air gas exchange.

- **Long-term trends:** Over the 14-year monitoring period, diffusive emissions of both $CH_4$ and $CO_2$ have declined. In contrast, $CH_4$ emissions from ebullition have remained relatively stable, sustained by the availability of OM pools in flooded sediments and vegetation.
    - In our next article, we calculated that the carbon inputs from the watershed, 34 GgC year$^{-1}$ on average, could have contributed to only 15% of total $CO_2$ and $CH_4$ emissions (Guérin, Deshmukh, Hoàng et al., to be submitted)

Those conclusions are more concise and emphasized well the key findings of the article.

In summary, we made several major changes as follows:

- Added two more co-authors: Nurholis Nurholis and Ari Putra Susanto.
- Added color to Fig. 2 and Fig. 4 (vertical profiles) to represent different seasons.
- Revised Abstract based on changes in the manuscript
- Revised Introduction with comments from reviewers
- Revised Material and Method:
    - Added information (STDEV) for windspeed and temperature in 2.1
    - Added year of min and year of max water level in 2.1
    - Added more information on Bubbling regarding to reviewer 2's comment in 2.7
    - Revised comparison EC – discrete sampling in 2.11, we used diffusion + ebullition (DE) instead of gross emission
    - Gross emission will not be put in abbreviation anymore.
    - Further QC/QA on EC results from 2019 and 2022 campaigns. Revised in 2.10.
- Revised Result:
    - Revised 3.1 on EC results
        - Added statistical tests for diurnal and comparison with DE (according to Reviewer 2's comment)
        - Revised flux values of $CH_4$ and $CO_2$ in Table 1 after further QC/QA and recalculated EC upscaling based on this change
    - Revised 3.2 on vertical profiles according to Reviewer 2's comment. The sub section was more concise.
- Revised Discussion:

o   Rearranged all the section as previously described

o   Added sub-section conclusions to every sub-section to condense the ideas

o   Revised diurnal variation and EC comparison sections as related to the new statistical tests and further QC/QA.

-   Revised Conclusion based on changes in the manuscript.

-   Added Data availability, Author contribution and Competing interests.

-   Revised References: based on Copernicus's template of reference.

**Below, we would like to answer to your point-to-point questions:**

**Line 55 and after: There are three mechanisms: ebullition, diffusive fluxes and degassing. One could briefly explain what these three are and how they are measured. Ebullitionà Bubble traps. Diffusive fluxes EddyCov, K\*DeltaC, ebullition upstream-downstream.**

*Answer*: I added the definition of each pathways only. The methods used to collect them were mentioned in the Material and methods (Section 2.2, 2.3)

**Line 88: The Dam was impounded and the commissioned.**

*Answer:* Yes, the impoundment of the reservoir started in 04/2008 and fully commissioned in 04/2010.

**Was happened when it was commissioned? Water through the turbines?**

Answer: Yes, Commissioning corresponds to the beginning of water turbining, which started in 03/2010

**Was the water before discharged via the spill-over?**

*Answer:* From 04/2008 to 03/2010, the water was discharged through the dams and spill-way (in the event of flood). After the turbines started, only 2 $m^3$ $s^{-1}$ flow was discharged regularly through the dams. The spillway was used only during flood events.

**Line 250: What are gross emissions. Is it already source and sink subtracted? Is it influx of OM minus GHG coming out?**

*Answer:* Gross emissions are those that are directly measurable from existing reservoirs (Rasanen et al., 2018), which means that it accounts only for the amount of emissions after the reservoir is stably impounded (2009 onwards), and from the water body only. The net emission of NT2,

considering also the emissions from the reservoir area before inundation, which can act as a GHG source (e.g. natural waters) or sink (e.g. forests) (Rasanen et al., 2018), which will be reported in a separate article with more pathways (fluxes from drawdown area and downstream emissions) and the downstream structures.

**Line 505-510: The ebullition effect of atmospheric pressure change. Did you see this also in the EC data?**

*Answer*: Yes, the EC measurement captures both diffusive fluxes and ebullitive fluxes from the water surface (Deshmukh et al., 2014)

**Line556: GE measurements (TBL and bubbling) is this the calculation method for Gross Emissions? This should be explained before.**

*Answer:* I added a term DE (diffusion + ebullition) instead of gross emissions for the comparison with EC to clarify. Also, I clarified the terminology for gross emission in Section 2.8

**Line 591: In the methods section there should be a clear definition of EC and TBL(GE) method. Then just use one abbreviation TBL or GE. I think that the way of calculating GE is through TBL. That should not be mixed up.**

*Answer*: I changed the term to DE when it comes to the comparison with EC.

**Line 608: kt values are often highly discussed and vary regarding which parametrization you use. This could be mentioned earlier when you compare the fluxes.**

*Answer:* the kt value is the important components of the calculation of diffusive fluxes. Since the first fluxes comparison presented in the diurnal variation only used the EC direct measurement data, hence the kt did not play an important role in this comparison. For $CH_4$ fluxes, it consisted of both bubbling and diffusion, and from the results showed in Section 3.3, in 2022, around 95% of the $CH_4$ rooted from bubbling. Hence the kt value is most significant when it comes to diffusive $CO_2$ fluxes which are the main source of $CO_2$ emissions and about 40% of the total GE.
We used both windspeed and rainfall effects to calculate kt value (TBL method)

**Line 617: "Temporal dynamics of CH4 emissions from the reservoir water surface." Why is the abstract called "from the water surface" you mention diffusive fluxes, ebullition and degassing and water discharged at the pill-over. What is so significant to the water surface now? You describe EC before and now the other pathways. I would choose a different subtitle.**

*Answer:* This paper strictly quantified the amount of $CO_2$ and $CH_4$ from the water body / water column of the main reservoir NT2, disregarding the emissions from the drawdown area (soil) and the downstream emissions (after the turbines). I will give a clearer answer below. About the title, I have reorganized the whole Section 4 discussion to be clearer and easier to follow for the audiences.

**Line 631: Do you mean the water is discharge from the reservoir or coming from the surroundings into the reservoir?**

*Answer:* I referred to the amount of DOC and IC coming from the watershed (tributaries) into the reservoir during the warm season.

**Line 780: Degassing + Feature. It should be written that the relative minor role of degassing due to the features at the damn, suggests that future projects….**

*Answer:* I added the information related to the design of NT2 reservoir: "such as intake configuration or artificial mixing system, which introduces destratification and oxygenation, upstream of the turbines"

**Line 790: This is a very interesting paragraph putting your measurements in relation to others**.

*Answer:* Thank you very much for this comment.

**Line 824: Can you just briefly remind us what the design features are: intake depth, ventilation, … just one or two catchwords.**

*Answer:* I added the information: "the introduction of artificial mixing of the water column before the turbines"

**General question: Did you take a deeper look at water-level influence. You mention the hydrostatic pressure changes influencing ebullition but only cite Deshmukh 2014. Have you seen any influence from falling dry and resubmerged banks?**

*Answer:* This article focused mainly on the GHG emissions ($CO_2$ and $CH_4$) from the water body and the corresponding pathways such as diffusion, ebullition, and degassing directly from the main reservoir (turbines, dam). The paper focused mostly on the processes underlaying the fluxes, which will contribute and compliment to another upcoming paper by our team reporting the broader view of net emissions which includes drawdown emissions (from the bank with the water variation effect as reported by Serca et al., 2016), downstream emissions (from the downstream structure of the NT2 reservoir, as reported in Deshmukh et al., 2016), the pre- and post-impoundment net balance and life cycle analysis (LCA). The results and discussions of this article will be fundamental to the net emission paper, as well as another article later for nitrogen circle and $N_2O$ emissions. From this impressive 14-year database of measurements, our team will provide the comprehensive reports on long-term emissions as well as biogeochemistry processes underlying those emissions.

**Technical remarks:** *I completed all the minor changes and included them to the final versions.*
**Abstract Line 31: I would spell it out in the abstract "warm dry". The same goes for cold dry. They are not too long. Its perfectly fine in the main body** – Done.

**Line 43: At the first mentioning I would write greenhouse gas (GHG)** – Done.

**Line 470/ Figure 5: You use ebullition and bubbling. I would recommend only use ebullition. This also would be appropriate for figures.** – Done, I changed to "ebullition" to match with the figure legend.

**References**

Chanudet, V., Fabre, V., and van der Kaaij, T.: Application of a three-dimensional hydrodynamic model to the Nam Theun 2 Reservoir (Lao PDR), Journal of Great Lakes Research, 38, 260–269, https://doi.org/10.1016/j.jglr.2012.01.008, 2012.

Chanudet, V., Smits, J., Van Beek, J., Boderie, P., Guérin, F., Serça, D., Deshmukh, C., and Descloux, S.: Hydrodynamic and water quality 3D modelling of the Nam Theun 2 Reservoir (Lao

PDR): predictions and results of scenarios related to reservoir management, hydrometeorology and nutrient input, Hydroécol. Appl., 19, 87–118, https://doi.org/10.1051/hydro/2014009, 2016.

Descloux, S., Chanudet, V., Taquet, B., Rode, W., Guédant, P., Serça, D., Deshmukh, C., and Guerin, F.: Efficiency of the Nam Theun 2 hydraulic structures on water aeration and methane degassing, Hydroécol. Appl., 19, 63–86, https://doi.org/10.1051/hydro/2015002, 2016a.

Descloux, S., Guedant, P., Phommachanh, D., and Luthi, R.: Main features of the Nam Theun 2 hydroelectric project (Lao PDR) and the associated environmental monitoring programmes, Hydroécol. Appl., 19, 5–25, https://doi.org/10.1051/hydro/2014005, 2016b.

Deshmukh, C., Serça, D., Delon, C., Tardif, R., Demarty, M., Jarnot, C., Meyerfeld, Y., Chanudet, V., Guédant, P., Rode, W., Descloux, S., and Guérin, F.: Physical controls on CH4 emissions from a newly flooded subtropical freshwater hydroelectric reservoir: Nam Theun 2, Biogeosciences, 11, 4251–4269, https://doi.org/10.5194/bg-11-4251-2014, 2014.

Deshmukh, C., Guérin, F., Labat, D., Pighini, S., Vongkhamsao, A., Guédant, P., Rode, W., Godon, A., Chanudet, V., Descloux, S., and Serça, D.: Low methane (CH4) emissions downstream of a monomictic subtropical hydroelectric reservoir (Nam Theun 2, Lao PDR), Biogeosciences, 13, 1919–1932, https://doi.org/10.5194/bg-13-1919-2016, 2016.

Deshmukh, C., Guérin, F., Vongkhamsao, A., Pighini, S., Oudone, P., Sopraseuth, S., Godon, A., Rode, W., Guédant, P., Oliva, P., Audry, S., Zouiten, C., Galy-Lacaux, C., Robain, H., Ribolzi, O., Kansal, A., Chanudet, V., Descloux, S., and Serça, D.: Carbon dioxide emissions from the flat bottom and shallow Nam Theun 2 Reservoir: drawdown area as a neglected pathway to the atmosphere, Biogeosciences, 15, 1775–1794, https://doi.org/10.5194/bg-15-1775-2018, 2018.

Guérin, F., Deshmukh, C., Labat, D., Pighini, S., Vongkhamsao, A., Guédant, P., Rode, W., Godon, A., Chanudet, V., Descloux, S., and Serça, D.: Effect of sporadic destratification, seasonal overturn, and artificial mixing on CH4 emissions from a subtropical hydroelectric reservoir, Biogeosciences, 13, 3647–3663, https://doi.org/10.5194/bg-13-3647-2016, 2016.

Serça, D., Deshmukh, C., Pighini, S., Oudone, P., Vongkhamsao, A., Guédant, P., Rode, W., Godon, A., Chanudet, V., Descloux, S., and Guérin, F.: Nam Theun 2 Reservoir four years after commissioning: significance of drawdown methane emissions and other pathways, Hydroécol. Appl., 19, 119–146, https://doi.org/10.1051/hydro/2016001, 2016.

**Response to Reviewer 2**

**Manuscript title:** Temporal variability of greenhouse gas ($CH_4$ and $CO_2$) emissions in a subtropical hydroelectric reservoir: Nam Theun 2 (Lao PDR)
**Manuscript number:** EGUSPHERE-2025-3295
**Journal:** *Biogeosciences*

**Authors:** Anh-Thái Hoàng, Frédéric Guérin, Chandrashekhar Deshmukh, Axay Vongkhamsao, Saysoulinthone Sopraseuth, Vincent Chanudet, Stéphane Descloux, Nurholis Nurholis, Ari Putra Susant, Toan Vu Duc and Dominique Serça
* * *
**Introduction**

We sincerely thank Reviewer #2 for their careful evaluation of our manuscript and for recognizing the value of the long-term and seasonally resolved dataset from this hydroelectric reservoir. We appreciate the reviewer's constructive feedback regarding the clarity and focus of the Results and Discussion sections.

sFor the context, this article represents a continuation and advancement of previous research initiated during the early years following the impoundment of the NT2 reservoir, spanning from 2009 to 2012. Earlier works (Deshmukh et al., 2014, 2014, 2018; Guérin et al., 2016; Serça et al., 2016) laid the foundational understanding of GHG dynamics and water quality conditions (Chanudet et al., 2016) within the framework of an extensive monitoring project (Descloux et al., 2016). The primary objective of this process-based article was to provide a comprehensive assessment of gross GHG emissions ($CO_2$ and $CH_4$) with temporal variation and seasonality of GHGs from the NT2 reservoir water. Moreover, the 14-year dataset will also support two additional research contributions. First, it enabled the accurate quantification of net GHG emissions from the reservoir (considering pre-impoundment, and additional pathways such as downstream emission and drawdown area emissions), including $N_2O$ fluxes, an important addition, and the application of updated life-cycle analysis methods (Guérin et al, in prep). Second, it contributed to an original study focused on the nitrogen cycle within the NT2 reservoir, marking a significant expansion beyond previous works which centered on carbon dynamics ($CO_2$ and $CH_4$). Collectively, these forthcoming articles, together with the current assessment, aim to provide a comprehensive understanding of biogeochemical processes governing greenhouse gas emissions from tropical reservoirs.

We fully acknowledge the reviewer's valuable suggestions regarding the structure and focus of the *Discussion* (Section 4) and the need for a more integrative and comparative *Conclusion*.

In response, we have undertaken substantial revisions as follows:

- We reorganized the Chapter 4, *Discussion*, as follows:
  - 4.1 Methodological comparison: evaluating the differences between EC and manual discrete sampling
  - 4.2 Diurnal variation of fluxes from EC measurement
    - 4.2.1 $CH_4$ fluxes
    - 4.2.2 $CO_2$ fluxes
  - 4.3 Seasonal variation of GHG emissions
    - 4.3.1 $CH_4$ emission
    - 4.3.2 $CO_2$ emission
  - 4.4 Annual variation of GHG emissions
    - 4.4.1 $CH_4$ emission
    - 4.4.2 $CO_2$ emission
    - 4.4.3 Comparison with the Petit-Saut (French Guyana) Reservoir: Long-term trends in GHG emissions

By this new organization, we emphasized each point of discussion clearer with sub-section according to each GHG ($CH_4$ and $CO_2$), so that the audience can easily follow. Also, we removed section 4.5 as it overlapped with the interannual variation sections, hence we added the comparison with Petit-Saut to section 4.4 as a subsection (4.4.3).

- We improved our key findings as follows:
  - Diurnal variations:
    - $CH_4$: clear diurnal variation ($p < 0.05$) as daytime fluxes higher than nighttime fluxes
    - $CO_2$: In 2010 and 2011, there was no significant difference between daytime and nighttime $CO_2$ fluxes ($p = 0.84$ and $0.80$, respectively). In contrast, in 2009 nighttime fluxes were significantly higher than daytime fluxes (Night > Day, $p < 0.05$). Thus, a clear diurnal variation in $CO_2$ was observed only during the WD season in 2009, when measurements were conducted in May. The 2010 and 2011 campaigns took place in March, during the late CD and early WD seasons, when no significant diurnal pattern was detected.

- While our estimates were considered robust, they should be interpreted as conservative due to the uncorrected diurnal variability.
  - Comparison of methods:
    - EC measurements provided continuous, high-temporal-resolution data, allowing capture of short-term events (for example, peaks in flux), process studies such as the impacts of changing water level or radiation on fluxes; and diurnal cycles, including nighttime emissions.
    - In contrast, discrete sampling captured spatial variability across the reservoir, but was limited to daytime measurements.
    - In this study, the discrete sampling dataset was considered more representative for overall emission calculations because it provided denser spatial coverage and higher data availability across the reservoir. Therefore, emission estimates reported in the article were primarily based on the discrete sampling approach, while EC data were used as complementary to support temporal dynamics and highlight short-term variability.
  - Emissions:
    - $CH_4$ and $CO_2$ fluxes exhibited distinct seasonal patterns linked to their dominant emission pathways:
      - $CH_4$ emissions peaked during the WD season, primarily driven by ebullition
      - $CO_2$ emissions peak during the CD season, dominated by diffusive fluxes associated with water column overturn, which brings $CO_2$-rich deeper water to the surface.
      - Drivers of GHG fluxes: Seasonal and short-term variations in $CH_4$ and $CO_2$ emissions are controlled by a combination of reservoir stratification, hydrological dynamics (e.g., water level fluctuations), and meteorological factors (e.g., temperature, wind, and precipitation). These factors influence the strength of diffusive fluxes, bubble formation, and water–air gas exchange.
      - Long-term trends: Over the 14-year monitoring period, diffusive emissions of both $CH_4$ and $CO_2$ have declined. In contrast, $CH_4$

emissions from ebullition have remained relatively stable, sustained by the availability of OM pools in flooded sediments and vegetation.

- o In our next article, we calculated that the carbon inputs from the watershed, 34 GgC year$^{-1}$ on average, could have contributed to only 15% of total $CO_2$ and $CH_4$ emissions (Guérin, Deshmukh, Hoàng et al., to be submitted)

Those conclusions are more concise and emphasized well the key findings of the article.

In summary, we made several major changes as follows:

- Added two more co-authors: Nurholis Nurholis and Ari Putra Susanto.
- Added color to Fig. 2 and Fig. 4 (vertical profiles) to represent different seasons.
- Revised Abstract based on changes in the manuscript
- Revised Introduction with comments from reviewers
- Revised Material and Method:
  - o Added information (STDEV) for windspeed and temperature in 2.1
  - o Added year of min and year of max water level in 2.1
  - o Added more information on Bubbling regarding to reviewer 2's comment in 2.7
  - o Revised comparison EC – discrete sampling in 2.11, we used diffusion + ebullition (DE) instead of gross emission
  - o Gross emission will not be put in abbreviation anymore.
  - o Further QC/QA on EC results from 2019 and 2022 campaigns. Revised in 2.10.
- Revised Result:
  - o Revised 3.1 on EC results
    - ▪ Added statistical tests for diurnal and comparison with DE (according to Reviewer 2's comment)
    - ▪ Revised flux values of $CH_4$ and $CO_2$ in Table 1 after further QC/QA and recalculated EC upscaling based on this change
  - o Revised 3.2 on vertical profiles according to Reviewer 2's comment. The sub section was more concise.
- Revised Discussion:
  - o Rearranged all the section as previously described
  - o Added sub-section conclusions to every sub-section to condense the ideas

       o   Revised diurnal variation and EC comparison sections as related to the new statistical tests and further QC/QA.

-   Revised Conclusion based on changes in the manuscript.

-   Added Data availability, Author contribution and Competing interests.

-   Revised References: based on Copernicus's template of reference.

**Below, we would like to answer to your point-to-point questions:**

- **Presentation of results and discussion – Much of the Results and Discussion are very long and detailed, and distract from the most important findings of the paper. For example, Section 3.2 in the Results takes up a lot of space for being only ancillary to the main findings of this paper. I suggest significantly reducing this section and perhaps putting it at the beginning of the Results (details could go in supplementary materials). The Results could then begin with a very brief overview of these vertical chemical dynamics, and follow with emphasis on the emissions comparisons between methods and over time, which are the real highlights of this paper. Similarly, the Discussion is very long winded, with many sections reading like a laundry list of similar studies without clearly linking to key findings of this study. I suggest focusing on linking specific results with mechanisms and relevant papers, and highlighting the specific, novel contributions from this study. Some closing paragraphs for long sections (i.e., paragraphs beginning on line 699 and 766) could succinctly summarize much of the section without needing multiple pages of text.**

*Answer:* thank you for this suggestion. The *Result* sections, especially the 3.2 about vertical profiles are essential for our theme of research in emissions of the NT2 reservoir. This article served as a process-based article to publish results that will be linked to a net-emission and carbon transfer article (Guérin et al., to be submitted). Hence, we would like to present the trends and values of each carbon species in details, together with mixing indicators (water temperature and dissolved oxygen). We revised this subsection to be more concise with the details.

The *Discussion* was reorganized and reduced as mentioned above with more precise conclusions for each section.

- **Integrating statistical tests – Much of the Results section uses visual or tabular comparisons to make statements about differences in emissions rates, when statistical tests should**

**explicitly be used. For example, comparisons of GE versus EC emissions are only presented using mean and uncertainty, when t-tests could easily compare these rates across all years. The authors write in line 349 emissions rate from EC were significantly higher than GE in some campaigns, including in March 2011 as 1.31 ± 0.10 Gg CH4 month-1 (EC) versus 1.25 ± 0.23 Gg CH4 month-1 (GE). However, considering the uncertainty values, these are not really different than each other. Please incorporate appropriate statistical tests for these and other comparisons (i.e., statistical trend tests for results in lines 486-492; mechanistic statement in line 620) rather than only qualitatively or visually describing results.**

*Answer:* We performed additional statistical tests on diurnal variation and EC comparison. For the prior results, only $CH_4$ showed clear diurnal variation (p<0.05, Night < Day in terms of flux magnitudes), while $CO_2$ showed diurnal variation only in the WD seasons (2009, p<0.05, Night > Day in term of flux magnitudes), while March 2010 and March 2011 showed no differences between Day and Night fluxes (p=0.84 and 0.80, respectively).

For the comparison, all of $CH_4$ comparison were significantly different (p <0.05), almost the same statistical test results showed for $CO_2$, only 2011 campaigns showed similarity between the two (p = 0.47).

We have done statistical tests on all emission results (trend; seasonal, interannual variations). We will incorporate those statistical results to support our claims, such as line 486-492 (p<0.05).

- **Upscaling from few stations to entire reservoir – The 3-7 sites used for upscaling emissions rates to the whole-reservoir scale seem very few to precisely capture the full range in spatial variability in emissions rates. How does this number of sites compare to recommendations in Beaulieu et al. 2016 (https://doi.org/10.1002/lno.10284, see Fig. 5) or to the area covered by each sampling site in previous work (i.e., Jager et al. 2022, https://doi.org/10.1016/j.rser.2022.112408, see Fig. 6)? The few numbers of sites is a major caveat for this scale of whole-reservoir upscaling and needs more attention and/or Discussion.**

*Answer:* The Nam Theun 2 reservoir hydrological assessment was published by Chanudet et al., 2012 (doi:10.1016/j.jglr.2012.01.008) mentioned due to their special hydrological conditions that RES9 is accounted for 3 km² before the turbine, while RES3 is accounted for 5.5% of the total area regardless of the seasonal fluctuation of water level.

The other 7 stations (RES1, RES2, RES4-8) were statistically tested for diffusive fluxes ($CH_4$ and $CO_2$) to see if they were spatially different. The results showed that they were statistically similar for

all 14 years of measurement. Hence, they represented the rest of the reservoir surface (100% - 5.5% - 3 km$^2$). These results were similar to those which have been reported by Deshmukh et al., (2014, 2018) and Guérin et al. (2016) in NT2 reservoir during the period 2009-2013. We included this information in the Section 2.5 Total diffusive fluxes calculation as special spatial variability test.

- **Linking proposed mechanisms to observed patterns – There are several instances in the Discussion where mechanisms are briefly proposed, but have limited statistical backing or additional explanations that are omitted. For example, in line 711, the measured $CO_2$ concentrations represent the net remaining between $CO_2$ consumption and uptake/emissions, not specifically the total $CO_2$ Hence, the production of $CO_2$ in the surface waters is likely more readily respired or emitted resulting in lower $CO_2$ concentrations, which is not specifically a reflection of the gross $CO_2$ production rates at depth. Another example in line 781 omits the accumulation of gases in deep water during the stratified period as a direct connection with higher degassing emissions during the warm stratified period. In general, more thorough mechanistic linkages would be warranted while removing much of the text that is not as relevant to the key findings of this study.**

*Answer:* Thank you very much for this suggestion.

Line 711: We changed from consumed by bacteria to consumed by phytoplankton as phytoplankton biomass increased during the WD season.

Line 781: In Nam Theun 2, degassing was reduced significantly due to the design of this reservoir. Before the turbine (RES9), there is an artificial mixing that outgassed both gases before downstream emissions. Before the dam (RES1), even though deep GHG-rich water going through the dam, but the discharge was only 2 m$^3$ s$^{-1}$ constantly. The high fluxes came only in the event of flood (spill-way release) but those events were rare.

Minor comments:

- **Line 56 – Disregarding degassing emissions can reduce emissions by a large portion (see, for example, Harrison et al. 2021, https://doi.org/10.1029/2020GB006888). Since you calculate and present degassing emissions, I suggest incorporating this into this rough breakdown of emissions by pathway.**

*Answer:* In addition to the previous remark on the degassing emissions, in Nam Theun 2 degassing only consisted of 4% of total emissions, and was stable through 14 years of study ($p > 0.05$). The

degassing process of NT2 was previously reported as minor pathway in Deshmukh et al., 2016 (doi:10.5194/bg-13-1919-2016) for $CH_4$ and will be reported again in Guérin et al., (to be submitted) for $CH_4$, and $CO_2$.

- **Line 58 – Consider here (and elsewhere as relevant) to discuss potential $CO_2$ uptake (i.e., $CO_2$ influx into the reservoir rather than emissions) that can be common in productive systems, as well as the variability in uptake versus emissions at diurnal to seasonal scales. This is particularly important given the negative $CO_2$ rates presented in Table 1.**

*Answer:* We had calculated the discharge of carbon and nutrient into the reservoir, and will be presented in next articles by Guérin et al., (to be submitted) on Net emissions and nitrogen cycling. The result showed that the contribution of the influx discharge to the reservoir contributed to only 15% of the total emissions.

Regarding the uptake from the surface (photosynthesis from phytoplankton for example), we did not have a situation where such conditions were observed in NT2 which is oligotrophic.

- **Line 75 – An eddy covariance flux tower is unlikely to be particularly useful for capturing spatial variability, as it integrates signal across a specific area without distinguishing variability in those signals within that area. Many would need to be set up around a reservoir to really get at spatial variability effectively. Given the scope of this paper with one EC tower set up, I suggest reducing the emphasis on capturing spatial variability via this method.**

*Answer:* Yes, we emphasized the EC continuous measurements are able to capture diurnal variation (daytime and nighttime fluxes) and short-event (peaks) in the discussion. Hence, we will rephrase this line to temporal variation only.

- **Line 146 – While ebullition does generally occur in shallower areas, it is possible to have ebullition occurring deeper than 16 m. I suggest rephrasing this and incorporating a couple of references that show ebullition more dominant – but not exclusive to – shallower areas in reservoirs (i.e., DelSontro et al. 2011, https://dx.doi.org/10.1021/es2005545; Beaulieu et al. 2016, https://doi.org/10.1002/lno.10284).**

*Answer:* In the NT2, we did not observe any $CH_4$ bubbling further than 16m of depth. This is case sensitive, therefore, we will rephrase it to specifically mention NT2.

- **Line 243 – Can you add more details on the ANN approach for the bubbling emissions estimates? This is a highly variable pathway over space and time, so additional details here (or in supplementary material) would be useful to describe model fitting and performance.**

*Answer:* In this study, an artificial neural network was used to find the best non-linear regression between ebullition fluxes and relevant environmental variables. We applied ANN using the 'nnet' package (https://cran.r-project.org/web/packages/nnet/index.html) to model bubbling fluxes. Water depth, change in water level, atmospheric pressure, change in atmospheric pressure, total static pressure, change in total static pressure, and reservoir bottom temperature data were used as explanatory variables - see the Deshmukh et al., 2014 for the choice of the inputs. The database of raw data was composed of 6,158 individual ebullition fluxes resulted from about 13 years of measurements (2009 – 2022), leading to a final input data for ANN composed of 6158 lines and 8 columns (1 output and 7 inputs). The data set is separated into two pools, the training one (80% out of total input data) and the validation one (20% out of total input data). The repeated cross-validation with 10 folds and 5 repetitions were applied to evaluate the performance of the ANN model. Additionally, the ANN model was iterated for 20 times. Averages of the 20 modelled values were used to estimate daily bubbling fluxes and the standard deviation was used to quantify the uncertainty in the estimates, the overall model performance reached up to 66% on the daily time scale.

I added this information to Section 2.7

- **Line 277 – Can you elaborate on this "spike removal process" for EC data processing? Given the often pulsed emissions nature of ebullition, it might be expected to find occasional to frequent "spikes" in CH$_4$ emissions as a result. Does this data processing remove those and hence artificially reduce the accuracy of ebullitive emissions variability?**

*Answer:* The "spike removal process" applied in our study followed a standard de-spiking procedure, in which both wind vector components and scalar exceeding three times the standard deviation from the local mean were flagged and removed.

This approach is widely used in EC studies to eliminate physically implausible or instrument-related outliers, while retaining the majority of real variability.

We acknowledge that CH$_4$ emissions from ebullition are often pulsed, and care must be taken to avoid artificially removing genuine high-flux events. In our implementation: The algorithm removes only extreme outliers that exceed the 3σ threshold, which primarily correspond to measurement artifacts or spikes unrelated to physical emission events. Short-term pulsed ebullition events that fall within the expected natural variability are retained, ensuring that the EC data continue to reflect the true temporal dynamics of CH$_4$ fluxes.

- **Line 320 – What specific results is this calculation of SD and SE used for? If used for whole-reservoir upscaled results (i.e., from the 3-7 measurement sites), consider utilizing propagated error based on individual measurement uncertainty and portion of the reservoir each represents.**

*Answer:* The 7 measurement sites represented an equal proportion of the reservoir after exclude RES9 and RES3. The SE was used to take into consideration the number of fluxes that made up the mean.

- **Line 338 – How comparable are the EC results from 2009-2011 versus 2019 and 2022, given the EC flux tower was (1) located in a very different part of the reservoir between these time periods, and (2) data span different months/seasons?**

*Answer:* We tested those results statistically, and they showed significant difference. This result are consistence with the finding that emissions declined with time for both gases $CH_4$ and $CO_2$ and they both showed distinct seasonal variations.

- **Line 341 – Suggest adding a figure (in the main text or supplementary) showing the differences between daytime and nighttime emissions patterns, as it is a major emphasis in the manuscript.**

*Answer:* Thank you for the suggestion. We added more information (statistical tests) to the diurnal fluxes.

- **Figure 3 – Suggest adding error bars to each estimate to visualize uncertainty (i.e., as in Figure 5), which would pair with explicit statistical tests as suggested above.**

*Answer:* Yes, I added the error bars to the two graphs.

- **Figure 4 – This figure is quite difficult to fully grasp. Consider reducing the number of panels/variables to include only those that are most different or relevant to the emissions patterns (moving other to supplemental material), and/or adding color/different line types to more clearly distinguish the different years. At its current size, the different symbols used are impossible to distinguish.**

*Answer:* After careful consideration, we chose to keep the figure as it is and the figure can be resize in the final version. The two main pathways (diffusion and ebullition) are clear, while degassing represents the gap in the middle.

**References**

Chanudet, V., Fabre, V., and van der Kaaij, T.: Application of a three-dimensional hydrodynamic model to the Nam Theun 2 Reservoir (Lao PDR), Journal of Great Lakes Research, 38, 260–269, https://doi.org/10.1016/j.jglr.2012.01.008, 2012.

Chanudet, V., Smits, J., Van Beek, J., Boderie, P., Guérin, F., Serça, D., Deshmukh, C., and Descloux, S.: Hydrodynamic and water quality 3D modelling of the Nam Theun 2 Reservoir (Lao PDR): predictions and results of scenarios related to reservoir management, hydrometeorology and nutrient input, Hydroécol. Appl., 19, 87–118, https://doi.org/10.1051/hydro/2014009, 2016.

Descloux, S., Chanudet, V., Taquet, B., Rode, W., Guédant, P., Serça, D., Deshmukh, C., and Guerin, F.: Efficiency of the Nam Theun 2 hydraulic structures on water aeration and methane degassing, Hydroécol. Appl., 19, 63–86, https://doi.org/10.1051/hydro/2015002, 2016a.

Descloux, S., Guedant, P., Phommachanh, D., and Luthi, R.: Main features of the Nam Theun 2 hydroelectric project (Lao PDR) and the associated environmental monitoring programmes, Hydroécol. Appl., 19, 5–25, https://doi.org/10.1051/hydro/2014005, 2016b.

Deshmukh, C., Serça, D., Delon, C., Tardif, R., Demarty, M., Jarnot, C., Meyerfeld, Y., Chanudet, V., Guédant, P., Rode, W., Descloux, S., and Guérin, F.: Physical controls on CH4 emissions from a newly flooded subtropical freshwater hydroelectric reservoir: Nam Theun 2, Biogeosciences, 11, 4251–4269, https://doi.org/10.5194/bg-11-4251-2014, 2014.

Deshmukh, C., Guérin, F., Labat, D., Pighini, S., Vongkhamsao, A., Guédant, P., Rode, W., Godon, A., Chanudet, V., Descloux, S., and Serça, D.: Low methane (CH4) emissions downstream of a monomictic subtropical hydroelectric reservoir (Nam Theun 2, Lao PDR), Biogeosciences, 13, 1919–1932, https://doi.org/10.5194/bg-13-1919-2016, 2016.

Deshmukh, C., Guérin, F., Vongkhamsao, A., Pighini, S., Oudone, P., Sopraseuth, S., Godon, A., Rode, W., Guédant, P., Oliva, P., Audry, S., Zouiten, C., Galy-Lacaux, C., Robain, H., Ribolzi, O., Kansal, A., Chanudet, V., Descloux, S., and Serça, D.: Carbon dioxide emissions from the flat

bottom and shallow Nam Theun 2 Reservoir: drawdown area as a neglected pathway to the atmosphere, Biogeosciences, 15, 1775–1794, https://doi.org/10.5194/bg-15-1775-2018, 2018.

Guérin, F., Deshmukh, C., Labat, D., Pighini, S., Vongkhamsao, A., Guédant, P., Rode, W., Godon, A., Chanudet, V., Descloux, S., and Serça, D.: Effect of sporadic destratification, seasonal overturn, and artificial mixing on CH4 emissions from a subtropical hydroelectric reservoir, Biogeosciences, 13, 3647–3663, https://doi.org/10.5194/bg-13-3647-2016, 2016.

Serça, D., Deshmukh, C., Pighini, S., Oudone, P., Vongkhamsao, A., Guédant, P., Rode, W., Godon, A., Chanudet, V., Descloux, S., and Guérin, F.: Nam Theun 2 Reservoir four years after commissioning: significance of drawdown methane emissions and other pathways, Hydroécol. Appl., 19, 119–146, https://doi.org/10.1051/hydro/2016001, 2016.